# Concurrent and orthogonal gold(I) and ruthenium (II) catalysis inside living cells

Cristian Vidal [1], María Tomás-Gamasa [1], Paolo Destito[1], Fernando López[1,2] & José L. Mascareñas [1]

The viability of building artificial metabolic pathways within a cell will depend on our ability to design biocompatible and orthogonal catalysts capable of achieving non-natural transformations. In this context, transition metal complexes offer unique possibilities to develop catalytic reactions that do not occur in nature. However, translating the potential of metal catalysts to living cells poses numerous challenges associated to their biocompatibility, and their stability and reactivity in crowded aqueous environments. Here we report a gold-mediated C–C bond formation that occurs in complex aqueous habitats, and demonstrate that the reaction can be translated to living mammalian cells. Key to the success of the process is the use of designed, water-activatable gold chloride complexes. Moreover, we demonstrate the viability of achieving the gold-promoted process in parallel with a ruthenium-mediated reaction, inside living cells, and in a bioorthogonal and mutually orthogonal manner.

[1] Departamento de Química Orgánica¸ Centro Singular de Investigación en Química Biolóxica e Materiais Moleculares (CIQUS), Universidade de Santiago de Compostela, Santiago de Compostela 15782, Spain. [2] Instituto de Química Orgánica General CSIC, Juan de la Cierva 3, Madrid 28006, Spain. These authors contributed equally: Cristian Vidal, María Tomás-Gamasa. Correspondence and requests for materials should be addressed to J.L.Mña. (email: joseluis.mascarenas@usc.es)

Nature has evolved a very complex cellular metabolism in which a myriad of enzymes works concurrently to catalyze multiple chemical reactions. Albeit yet no more than a dream, scientists might one day be able to build biocompatible, customized metabolic networks based on artificial catalysts and/or enzymes. Progress towards this goal requires the development of effective catalysts capable of achieving programmed and bioorthogonal transformations in the crowded environment of living cells.

In recent years, there has been an increasing number of reports on the application of transition metal-catalyzed reactions in biological settings and, in some cases, even in intracellular environments[1–6]. It is pertinent to note that while the term catalysis is commonly used, intracellular turnover has not been really investigated. Up to now, these reactions have been essentially restricted to the use of copper, palladium, and ruthenium complexes[7–16], while other important transition metals in organometallic catalysis, such as gold[17–19], have not been yet explored. Nevertheless, isolated reports on the detection of toxic Au(III) salts in biological media, which rely on gold-promoted transformations, suggest the viability of using gold catalysis in bio-relevant aqueous settings[20–25]. Tanaka et al. have very recently reported the use of a glycoalbumin-gold(III) complex for a propargyl ester amidation in mice;[26] curiously, control experiments in biological media and/or cultured cells were not described. A depropargylation reaction promoted by heterogeneous gold nanoparticles in living settings has been recently described[27].

The power of gold catalysis in synthetic chemistry stems, in great part, from the ability of gold cationic complexes to activate π-bonds in a chemoselective manner[28–30], and the possibility of tuning their reactivity by changing the electronic and steric characteristics of the ligands[31]. Furthermore, the processes promoted by gold complexes, especially by gold(I) species, tend to be tolerant to air and moisture. In many cases the reactions can be carried out using gold(I) chlorides, but they require the addition of chloride scavengers such as silver(I) salts to replace chloride by a more labile ligand. Curiously, some isolated reports on gold-promoted transformations in water, developed in the context of green chemistry, suggest that in this solvent such scavengers

might not be strictly needed[32–39]. This is particularly relevant when one envisions to translate the power of gold catalysis to biologically relevant aqueous environments, and eventually, to native cellular settings.

On these grounds, we reasoned that appropriately designed gold(I) chloride complexes with the structure [AuCl(L)] (L = ligand) might offer exceptional opportunities to design cell-compatible, bioorthogonal catalysts. The presence of the ligand might provide for the modulation of the reactivity, solubility, cell uptake and toxicity of the complex, and even allow for their conjugation to designed partners. On the other hand, the chloride ion ensures stability and an easy access to a variety of complexes (Fig. 1a), while eventually providing for a direct activation of the catalysts under aqueous conditions.

Herein we demonstrate that discrete gold(I) chloride complexes featuring designed water compatible ligands are highly efficient catalysts for achieving mild intramolecular alkyne hydroarylation in aqueous media (Fig. 1a). The reaction is highly bioorthogonal, and can be carried out in biological media, and even in intracellular environments, without raising significant toxicities. Importantly, we demonstrate the viability of conducting simultaneous gold and ruthenium-mediated reactions within living cells and without cross-reactivity. Albeit yet very simple, the system represents the first non-natural, mutually orthogonal, metal-promoted transformations capable of operating in the living environment of a mammalian cell (Fig. 1b).

## Results

**Gold-promoted carbocyclizations in aqueous media**. Discovering metal-promoted reactions taking place in complex aqueous environments requires an easy way of monitoring the transformation. In this context, the use of probes that increase their fluorescence upon undergoing the reaction is especially convenient. While several probes which undergo fluorescence-inducing heterocyclizations in presence of AuCl$_3$ have been described[40–43], we preferred to focus on more challenging carbocyclizations owing to the added synthetic significance of making carbon–carbon bonds[44]. Thus, at the outset, we chose the diethylamine precursor 1, which has been previously used for

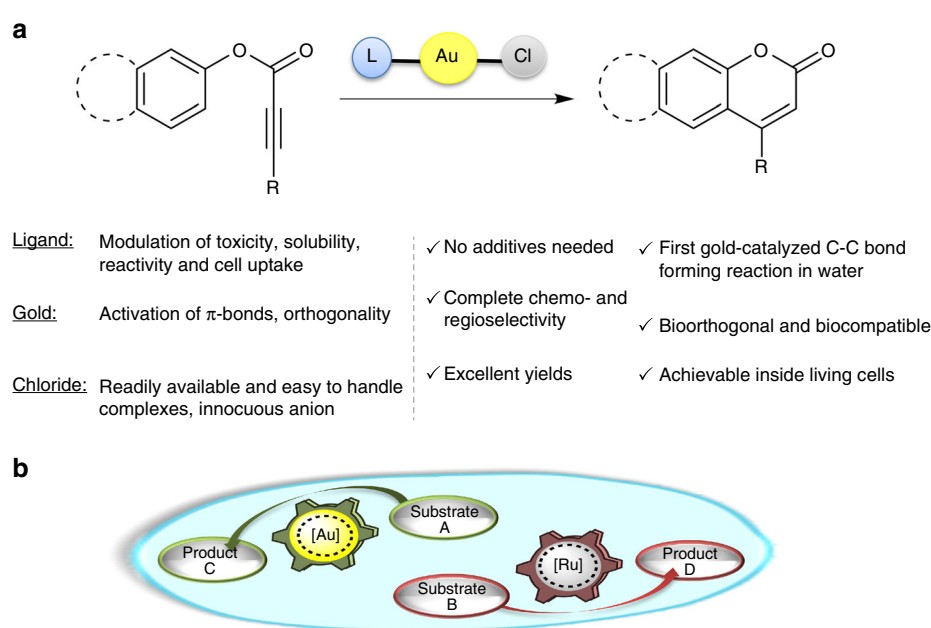

**Fig. 1** Au(I)-promoted cyclization and intracellular transformations. **a** Gold-chloride complexes as ligand-tuned, water-activatable precatalysts, and proposed gold-promoted carbocyclization. **b** Mutually orthogonal, and bioorthogonal, gold and ruthenium catalysis inside living cells

**Fig. 2** Intramolecular gold-catalyzed hydroarylation of precursor **1**. **a** Schematic representation of the carbocyclization reaction. **b** Structure of the gold complexes **Au1–Au6**, and gold salts **Au7–Au8** used in this study

gold-catalyzed hydroarylation reactions in organic solvents, but not in aqueous media (Fig. 2a)[22]. The reaction was explored in water, at 37 °C, using several Au(I) and Au(III) complexes (Fig. 2b).

We observed conversions higher than 70% with some of these gold complexes (see Supplementary Table 1), which confirm the viability of achieving the gold-catalyzed process in water, in absence of scavenging additives. The direct use of gold chloride catalysts in aqueous solvents has few precedents, none of them studying the activating role of water as a formal chloride scavenger[32–39]. To shed some light on this water-promoted activation process, we performed a series of NMR and ESI-MS studies (detailed in the Supplementary Note 7) which confirmed that water facilitates the ionization of the Au–Cl bond, and thereby triggers the reactivity of the gold complexes. Likely, both, the dielectric constant of water, and its ability to scavenge and solvate the chloride anion, are responsible for the ionization process.

Despite the success in hydroarylation of substrate **1**, the reaction provided mixtures of regioisomeric coumarins **2a** and **2b**, which somewhat complicates the analysis of the process[45]. Therefore, we moved to the procoumarin substrate **3**, in which one of the reactive positions is blocked, and thus produces a single regioisomer. Moreover, the reaction generates a highly fluorescent coumarine product **4**, with intense emission not only in the blue, but also in the green region (see Supplementary Figures 8–10).

The reactivity of this probe in different media was initially studied with gold complex **Au1** (10 mol%), which features a PTA ligand (PTA = 1,3,5-triaza-7-phosphaadamantane) (Table 1).

When water was used as the sole solvent we could only isolate the product with 12% of yield after 24 h at 37 °C (Table 1, entry 1). Given that this poor performance seemed to be associated to the limited water solubility of the substrate, we studied the process in the presence of different organic co-solvents. While we observed no conversion in a 1:1 mixture of water:toluene (entry 2) or water:methanol (entry 3), we were glad to obtain good yields of the product when the reactions were carried out in water: tetrahydrofuran or water:acetonitrile mixtures (entries 4 and 5, 71% and 99% yield, respectively). Importantly, the use of just acetonitrile led to almost complete recovery of the starting

material (entry 6), a result that is in agreement with the NMR studies, and the requirement of water to generate an active catalyst. As expected, the reactivity in acetonitrile could be efficiently recovered by adding AgOTf (10 mol%) to the mixture (entry 7). Finally, we found that with just 20% vol. of acetonitrile, it was possible to obtain the product in 83% isolated yield after only 3 h of reaction (Table 1, entries 9–10). Moreover, the catalytic loading could be reduced to 5 mol% without affecting the efficiency and yield of the process (entry 11), and even to 0.5 mol %, albeit in this case the reaction required heating at 75 °C (entry 12). Interestingly, when the reaction was carried out in a 6 M solution of NaCl instead of water (using 20% of acetonitrile as co-solvent), there was no conversion. This result is in consonance with the requirement of dissociation of the chloride anion for triggering the reactivity of the gold complex, and with the existence of an equilibrium between gold(I)-aqua and gold(I)-chloride species in water (Supplementary Note 7)[46].

**Performance of other complexes**. Once we found out the optimal conditions for the hydroarylation reaction of **3** with **Au1**, we tested other gold complexes (Fig. 2b). The water-soluble gold chloride complexes **Au2–Au5** behaved as well as **Au1**, promoting the formation of the product in excellent yields after 24 h at 37 °C. The gold(III) salts NaAuCl4 (**Au7**) and HAuCl4 (**Au8**) presented a lower efficiency (70% and 65% yield, respectively). When the reaction was carried out using [AuCl(PPh3)] (**Au6**), we observed no conversion, most probably because of the low aqueous solubility of the complex which hinders its water promoted activation (see the Supplementary Figure 20).

**Cellular viability**. Critical to the final goal of translating metal-catalyzed reactions to cellular habitats is the control of the toxicity. This is particularly relevant when dealing with gold complexes, since it has been shown that some of them can be quite cytotoxic[47, 48]. Therefore, we studied the viability of living human cervical cancer cells (HeLa cells) in presence of several concentrations of the different gold complexes using standard cytotoxicity tests (MTT and propidium iodide assays, see Fig. 3a and Supplementary Figure 21). Both the substrate **3** as well as the reaction product **4** present very low toxicity (around 5% of dead cells after 7 h at 75 μM). Gratifyingly, complex **Au1** showed the

**Table 1 Influence of the solvent and reaction conditions in the Au1-catalyzed hydroarylation of precursor 3**

| Entry[a] | Solvent | Yield | Entry[a] | Solvent | Yield |
|---|---|---|---|---|---|
| 1 | $H_2O$ | 12 | 7 | MeCN | 99[b] |
| 2 | $H_2O$/Toluene 1:1 | 0 | 8 | $H_2O$/MeCN 9:1 | 54 |
| 3 | $H_2O$/MeOH 1:1 | 0 | 9 | $H_2O$/MeCN 8:2 | 99/83[c] |
| 4 | $H_2O$/THF 1:1 | 71 | 10 | $H_2O$/MeCN 8:2 | 99[d]/82[c] |
| 5 | $H_2O$/MeCN 1:1 | 99 | 11[e] | $H_2O$/MeCN 8:2 | 99/82[c] |
| 6 | MeCN | 0 | 12[f] | $H_2O$/MeCN 8:2 | 99/81[c] |

[a]Reaction conditions: substrate **3** (0.05 mmol), **Au1** (10 mol%), solvent (total volume: 1.0 mL), 37 °C, 24 h
[b]AgOTf (10 mol%) was used as additive
[c]Yield of isolated product
[d]3 h
[e]5 mol%, 6.5 h
[f]0.5 mol%, 75 °C, 16 h

lowest cytotoxicity among the eight complexes so far tested, with a cell viability of 100% at 25 µM after 6 h of incubation, while under the same conditions (25 µM), **Au2**[49] led to a decrease in the viability of 20%. Complexes with the carboxylate, **Au3**, and specially, sulfonate appendages, **Au4**, were considerably more toxic. These results corroborate the role of the ligands to tune the cytotoxicity of the gold complexes.

**Bioorthogonality**. In addition to exhibiting good reaction rates in aqueous solvents (high yields in less than 2–3 h) and low toxicities under physiological conditions, a critical element to use metal promoted transformations in biological settings is their bioorthogonality. Gratifyingly, our gold-catalyzed hydroarylation of **3** using **Au1** proceeds very efficiently in the presence of one equivalent (with respect to the substrate) of carbohydrates such as glucose, or different amino acids like glycine (aliphatic), glutamic acid (acid), lysine (basic), tyrosine (aromatic), valine (essential) or methionine (sulfur containing) (see Supplementary Figure 1). Not surprisingly, the presence of excess of thiols (glutathione and cysteine) in the reaction media led to an important inhibition of the catalytic activity, likely due to the competitive coordination of the thiols to the gold. One equivalent of adenine, cytosine, and histidine also had a poisoning effect, but when using only 10 mol % (one equivalent with respect to the gold complex) we observed very good conversions (71% and 94% of yield in presence of histidine and cytosine, respectively). Remarkably, the reaction preserved its excellent selectivity to give the hydroarylation product **4**, as potentially competing products resulting from the intermolecular addition of these nucleophilic bases to **3** were never detected.

This reasonable biorthogonal profile prompted us to study the transformation in biological media of diverse complexity. We were pleased to find that the experiments proceeded with excellent yields, above 90%, when performed in PBS (phosphate buffered saline solution 1×, pH = 7.4) and DMEM (Dulbecco's modified Eagle's medium in which the cells are grown). The reaction is also compatible with the presence of proteins like BSA (bovine serum albumin protein), which features one free cysteine, and several histidines in their structure. More importantly, the reaction also takes place in the presence of cell lysates, albeit with lower yields (27%). Finally, the transformation can also be carried out in presence of living bacteria (see Supplementary Table 3).

**Intracellular reactions**. While the above results confirm that synthetically relevant gold-catalyzed processes can be efficiently achieved in complex aqueous mixtures, the translation to the interior of living cells poses major, additional challenges derived from a highly crowded and compartmentalized milieu, and the requirement of an efficient cell entrance and diffusion of the probes and reagents.

Control experiments revealed that addition of coumarin **4** (100 µM) to HeLa cells, followed by double washing with DMEM, elicits a clear intracellular blue and green fluorescence. Interestingly, while the blue signal presents a more vesicular distribution, the green fluorescence is extended across the cytoplasm. In contrast, cells treated with the substrate **3** were essentially non-fluorescent (see Supplementary Figure 22).

The reactivity tests were carried out by incubating the cells with the gold complexes (50 µM) for 30 min in fresh DMEM. After two washing steps with DMEM, the resulting cells were incubated with the substrate **3** (100 µM) in fresh DMEM for different reaction times, and washed twice with DMEM. Fluorescence should only develop inside the cells containing both the substrate and the catalyst. As shown in Fig. 3, using the gold complex **Au1** we were glad to observe a very clear fluorescence build-up inside the cells, consistent with the formation of coumarin product **4** (Fig. 3b, panels A and B and enlarged image in Fig. 3c, panels B and C). After 6 h, the blue and green emissions present a good co-localization, with a quite vesicular pattern (Fig. 3b, panel C and enlarged image in Fig. 3c, panel F). Interestingly, control incubations with a lysosomal marker like LysoTracker red (100 nM), allowed to observe an excellent overlay with the green and the blue fluorescence emissions (Mander's coefficients ~90%[50], Fig. 3c, panels D and E, respectively), suggesting that the product accumulates in lysosomes.

Monitoring the process by fluorescence microscopy at different times revealed changes in the intracellular distribution of the product. As shown in the Supplementary Figure 23, after 2 h of addition of **3** the cells displayed a diffuse fluorescence, while after 6 h the signal was concentrated in vesicles.

Flow cytometry analysis confirmed that HeLa cells incubated just with substrate **3** (Fig. 4a, pale purple) are essentially non-fluorescent (Green-B emission filter, 512/18 nm). However, standard incubation of the cells with the gold complex **Au1**, washing, and addition of the precursor **3**, led to a huge increase in

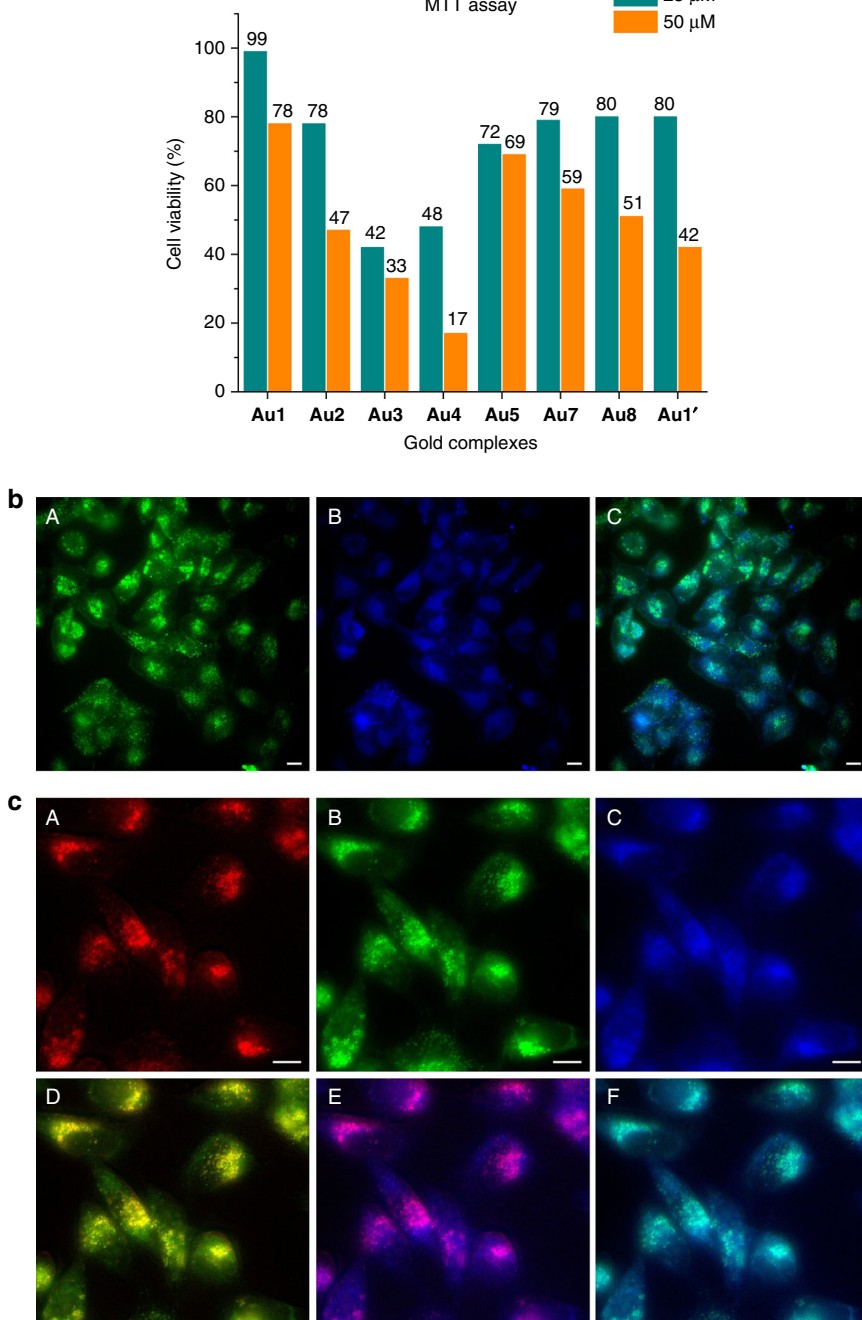

**Fig. 3** Reactivity of gold complex **Au1** in HeLa cells. **a** Bars representation of the cell viability. HeLa cells were incubated in cell culture medium containing the indicated amounts of the gold complexes for 6 h, and the amount of viable cells was analyzed using an MTT assay. The viability is expressed as the fold change of the fluorescence/absorbance value with respect to untreated cells (value 1.0). Error in these measures was less than 6% (3 replicates). **b** (A) Fluorescence micrographies of cells incubated with the gold catalyst **Au1** followed by addition of substrate **3** (green channel); (B) Similar experiment (blue channel); (C) Merging of A, B. **c** Subcellular localization of the catalytic activity of **Au1**. (A) Lysosomal labeling with Lysotracker (red); (B) Fluorescence micrographies of cells incubated with the gold catalyst **Au1** followed by addition of substrate **3** (green channel); (C) Similar experiment (blue channel); (D) Merging of A,B; (E) Merging of A and C; (F) Merging of B, C. Reaction conditions: Cells were incubated in DMEM with the gold complex (50 µM in DMSO) for 30 min, followed by two washings with DMEM and treatment with substrate **3** (100 µM) for 6 h. When using Lysotracker cells were incubated with 100 nM of this marker for 15 min. Scale bar: 12.5 µm

fluorescent cells after 6 h (Fig. 4a, pink). Complex **Au2** was also a very good catalyst, and **Au5** was also able to elicit the intracellular transformation of **3**, but with a poorer performance than **Au1** and **Au2** (Fig. 4b), while NaAuCl₄ (**Au7**) failed to promote the desired

reaction. Complexes **Au3** and **Au4** and tetrachloride salt **Au8** were not selected for the study due to their relative high toxicity. Interestingly, a control experiment with a preformed complex similar to **Au1** but bearing a bistriflimidate [NTf₂] instead of a

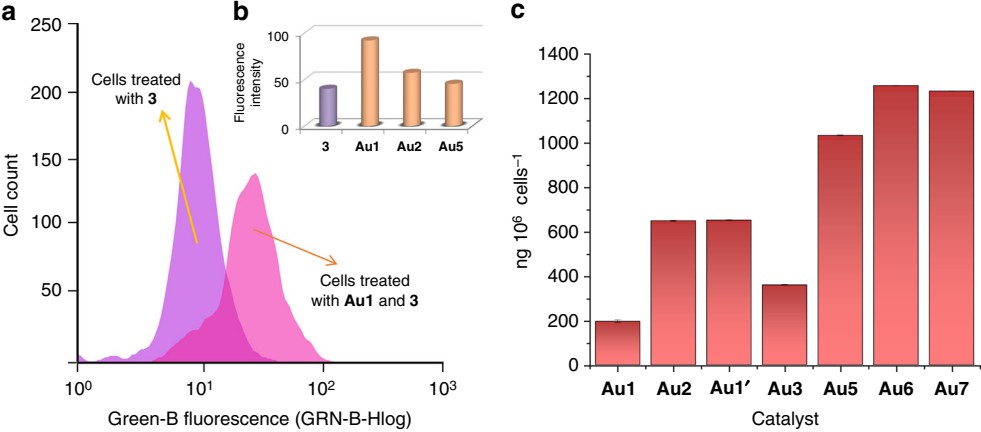

**Fig. 4** Flow cytometry and ICP studies with gold complexes in HeLa cells. **a** Histogram showing a population of cells treated with substrate (pale purple) versus cells pretreated with **Au1** and then incubated with substrate **3** (pink). **b** Quantification of fluorescent cells after the reactions promoted either by **Au1**, **Au2** or **Au5** (pink bars). Reaction conditions: Cells were incubated with the gold complex (50 μM) for 30 min, washed with DMEM and treated with substrate **3** (100 μM) for 6 h. **c** ICP-MS results on the intracellular accumulation of gold after incubation of cells in DMEM with 75 μM of gold complexes (in DMSO) for 2 h, double washing with PBS and digestion with $HNO_3$

chloride ligand ([AuNTf$_2$(PTA)], **Au1'**), revealed that this catalyst was much less efficient in cellular media (see Supplementary Figure 24). This result is especially noticeable, as bistriflimidate-gold(I) complexes tend to be excellent catalysts in standard solvents;[13] and further supports the relevance of using the chloride-based complexes (e.g., **Au1**) for reactions in biological milieu. Moreover, complex **Au1'** turned out to be slightly more toxic than its corresponding chloride counterpart **Au1** (Fig. 3a and Supplementary Figure 21).

To further confirm that the reactions involve a gold-catalyzed process, we measured the presence of gold inside the cells by ICP-MS analysis of samples prepared by incubation with the complexes, followed by two thorough washing steps with PBS, and digestion with $HNO_3$. As shown in Fig. 4c, all complexes led to accumulation of gold inside cells, but the uptake was different depending on the ligand. With complex **Au1** we observed a gold amount of 195.89 ng 10$^6$ cells$^{-1}$, and with **Au2** the amount of gold was three times higher than with **Au1**. Gold(III) chlorides **Au5** and **Au7** and phosphine derivative **Au6** led to a higher accumulation of gold, however they were less effective catalysts than **Au1** or **Au2**. These results confirm that gold species can effectively enter living cells, but also point out that the entrance depends on the characteristics of the ligand, and that there is not a necessary correlation between internalization and catalytic performances.

**Concurrent and orthogonal Au/Ru intracellular reactions**. Having demonstrated the viability of performing a gold-promoted hydroarylation inside living cells, we wondered whether this gold-mediated process could be achieved in parallel with other intracellular metal catalyzed transformations. Having several metal catalysts operating in a concurrent, chemoselective, orthogonal, and bioorthogonal manner inside living cells, represents an appealing, challenging goal, that has not been previously achieved. The viability of the approach was explored using as second transformation, a deallylation reaction promoted by a ruthenium complex[8].

In particular, we chose the reaction indicated in Fig. 5a, in which the deallylated product **6** presents an infrared fluorescence, and does not interfere with the green/blue fluorescence of the product **4** generated in the gold(I) mediated reaction. We selected the triphenylphosphonium-ruthenium derivative **Ru1** as catalyst,

as it is known to be efficiently uptaken by the cells and doesn't present intrinsic fluorescence.

Remarkably, addition of substrates **3** (100 μM) and **5** (50 μM) to cells that had been previously treated with complexes **Au1** (50 μM) and **Ru1** (25 μM) and thoroughly washed with DMEM, led to an intense intracellular infrared fluorescence coming from product **6** (Fig. 5b, panel D) and typical green and blue staining patterns of product **4** (Fig. 5b, panels H and L). Cross experiments confirmed that neither **Ru1** was able to promote the gold-catalyzed reaction, nor **Au1** catalyzed the ruthenium transformation (See Supplementary Figure 25 and 26). Indeed, infrared fluorescence was only generated after incubation of **Ru1** with substrate **5** (Fig. 5b, panels B, F, and J), while the green and blue emission were observed in cells treated with **Au1** and substrate **3** (Fig. 5b, panels C, G, and K). These results represent the only example of an orthogonal and bioorthogonal dual metal catalyzed reaction inside living cells.

In conclusion, we have demonstrated that readily available and ligand-tunable gold(I)-chloride complexes can promote efficient hydroarylation processes in complex aqueous media and even inside living mammalian cells. In contrast to standard gold(I) catalysis in organic solvents, the reactions do not require the addition of chloride scavengers, something critical when considering the translation of this chemistry to biological environments. Our data suggest that water, a necessary solvent in such environments, is somewhat working as catalyst-activating agent. The reaction, which involves a carbon–carbon bond cyclization in water using a gold catalyst, is highly bioorthogonal, biocompatible, and can be efficiently carried out inside living cells.

While the gold-promoted reaction represents a significant addition to the toolbox of life compatible transformations, we have also demonstrated that it can be achieved inside cells in parallel with another metal-promoted process, namely, a ruthenium promoted deallylation. The ability to run multiple, designed catalytic transformations within cells might open novel avenues for biological intervention and is particularly intriguing from the more fundamental perspective of engineering metal-based, fully artificial metabolic networks.

## Methods

**General**. Chemical synthesis procedures, detailed protocols, and characterization of all the compounds are included in the Supplementary Information. For NMR,

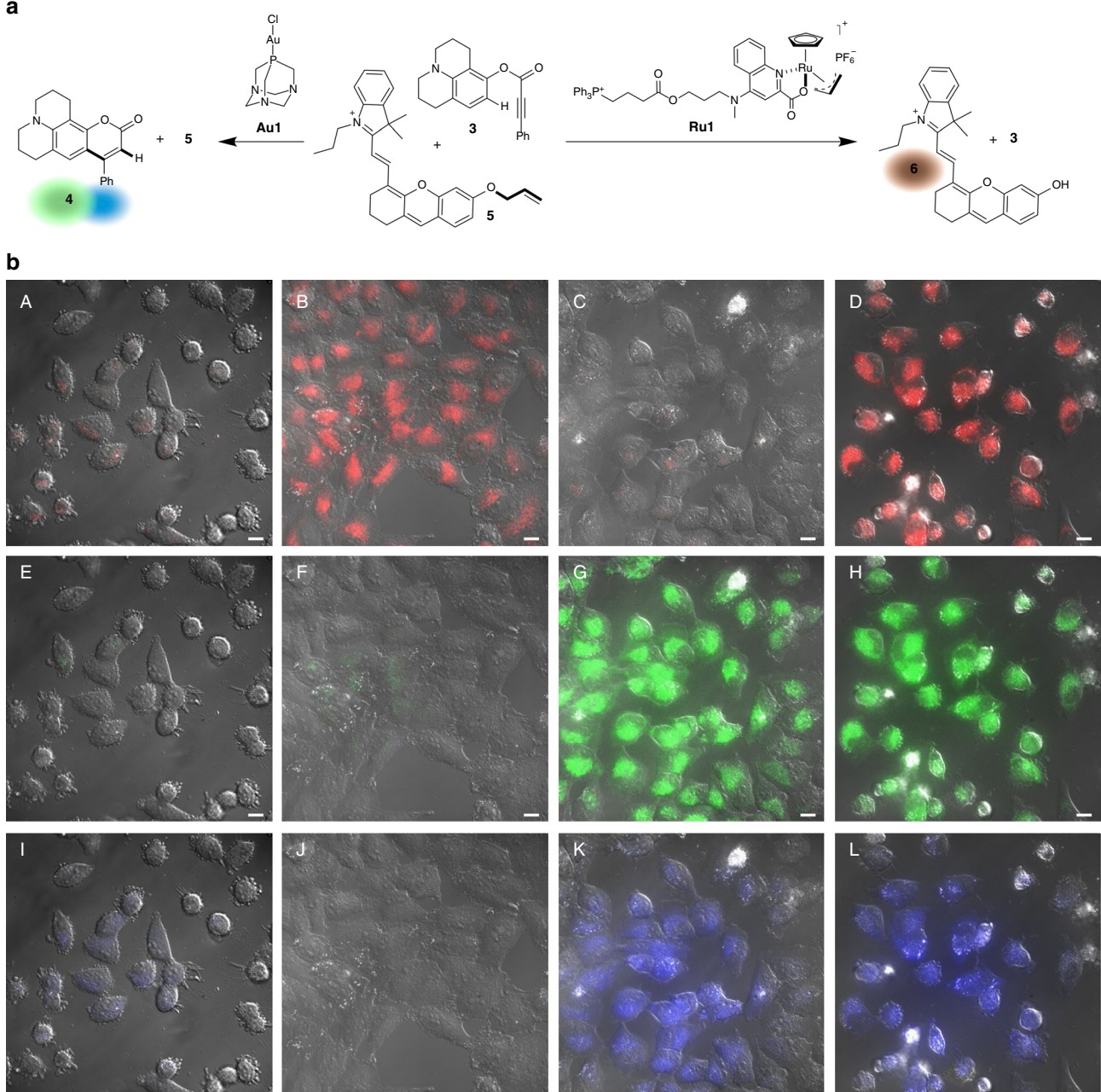

**Fig. 5** Concurrent Au(I) and Ru(II) catalysis inside HeLa cells. **a** Schematic representation of the ruthenium and the gold-mediated reactions. **b** Intracellular transformations. A, E, I) Red, green, and blue fluorescence of the substrates **3** and **5** in cells; (B, F, J) Red, green, and blue fluorescence of cells incubated with **Ru1** followed by addition of substrate **5** and **3**; (C,G,K) Red, green, and blue fluorescence of cells incubated with **Au1** followed by addition of substrate **3** and **5**; (D, H, L) Red, green, and blue fluorescence of cells incubated with **Ru1** and **Au1** followed by addition of substrates **3** and **5**. Reaction conditions: Cells were incubated with **Ru1** (25 μM) and/or **Au1** (50 μM) for 30 min, followed by two washings with DMEM and treatment with substrate **3** (50 μM) and/or **5** (100 μM), respectively, for 6 h. Scale bar: 12.5 μm

UV, and fluorescence analysis of the compounds in this article, see Supplementary Figures.

**Synthesis of compound (3)**. Phenyl propiolic acid (5.495 mmol, 0.803 g, 1.3 eq.) and CH$_2$Cl$_2$ (6.4 mL) were successively added to a heat gun dried round bottom flask equipped with a stir bar under nitrogen. The mixture was stirred at 0 °C in an ice/brine bath for 1 min. Then, neat DIC (*N*,*N*-dicyclohexylcarbodiimide, 6.341 mmol, 0.993 mL, 1.5 eq.) was added via syringe and the mixture was stirred for 1 min until a white precipitate was formed. A solution of 8-hydroxyjulolidine (4.227 mmol, 0.80 g, 1.0 eq.) in CH$_2$Cl$_2$ (10.0 mL) was added via syringe followed by DMAP (4-(dimethylamino)pyridine, 1.057 mmol, 0.129 mg, 0.25 eq.) in CH$_2$Cl$_2$ (1.0 mL). The mixture was stirred at 0 °C until complete consumption of the phenol was observed by TLC. Then the reaction mixture was filtered through

Kieselguhr and concentrated in vacuum. The residue was purified by silica flash column chromatography using hexane/EtOAc (8:2) as eluent to yield the product **3** as a yellow/orange solid (2.830 mmol, 0.890 g, 67% yield).

**Representative procedure for the catalytic reaction in water**. Substrate **3** (0.050 mmol, 15.90 mg) was added to a Schlenk tube containing a stir bar followed by the addition of [AuCl(PTA)] (**Au1**, 0.005 mmol, 1.90 mg, 10.0 mol%). MeCN (200 μL) was added and the reaction mixture was stirred at 300 rpm until a homogenous solution was formed (10 s). H$_2$O (800 μL) was added (final volume 1.0 mL, [Substrate] = 50.0 mM), the Themowatch-controlled heating block was fixed at 37 °C and the reaction was stirred for 3 h. After this time, the reaction mixture was extracted with CH$_2$Cl$_2$ (3 × 10.0 mL) and the combined organic fractions were dried, concentrated over silica and purified by silica flash column

chromatography using hexane/EtOAc (8:2) as eluent to obtain the product **4** as a yellow solid (0.041 mmol, 13.17 mg, 83%).

**Cell culture experiments**. Cell lines were cultured in DMEM supplemented with 10% (v/v) fetal bovine serum (FBS), and containing 5 mM glutamine, penicillin (100 units mL$^{-1}$) and streptomycin (100 units mL$^{-1}$) (all from Invitrogen). Proliferating cell cultures were maintained in a humidified incubator at 37 °C and 5% CO$_2$ atmosphere. All incubations were performed in DMEM at 37 °C.

**Reactions in living cells**. Cells growing on glass coverslips were incubated with either gold complexes **Au1–Au8** (50 μM) for 30 min, washed twice with DMEM and incubated with substrate **3** (100 μM) for 6 h. The samples were washed twice with fresh DMEM and the coverslips observed in vivo in a fluorescence microscope equipped with adequate filters. Identical conditions of gain and exposure were applied for all the digital pictures of the different samples.

**ICP analysis**. For the ICP measurements, a total of $3 \times 10^6$ HeLa cells growing in six well plates were treated with 25–75 μM of the different gold complexes (**Au1–Au3, Au5–Au7**) in DMEM for 1 h. Prior to digestion, the samples were washed with fresh DMEM and then twice with PBS. The obtained fractions were digested in triplicate in HNO$_3$ by microwave heating and analyzed.

**Flow cytometry studies**. After the incubation time, cells were washed once with DMEM, then twice with PBS, harvested with trypsin for 15 min and resuspended in PBS buffer supplemented with 2% of FBS and 5 mM EDTA. The intracellular catalysis was analyzed by flow cytometry. Quantification was measured under the 512/18 nm band pass filter.

**Viability assays**. The toxicity of the gold complexes was tested by means of the propidium iodide and MTT assays in HeLa cell line.

**Data availability**. The authors declare that the data supporting the findings of this study are available within the article and its Supplementary Information files. Other data that support the findings of this study are available from the corresponding author upon request.

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

## Acknowledgements

This work has received financial support from Spanish grants (SAF2013-41943-R, SAF2016-76689-R, Orfeo-cinqa network CTQ2016-81797-REDC), the Xunta de Galicia (2015-CP082, ED431C 2017/19, and Centro Singular de Investigación de Galicia accreditation 2016-2019, ED431G/09), the European Union (European Regional Development Fund-ERDF), and the European Research Council (Advanced Grant No. 340055). M.T.G. thanks the Ministerio de Economía y Competitividad for the Juan de la Cierva-Incorporación fellowship (IJCI-2015-23210). The authors thank R. Menaya-Vargas for excellent technical assistance and M. Marcos for helpful contributions on MS analysis.

## Author contributions

C.V. and M.T.G. contributed equally to this work. C.V. and P.D. performed the chemical synthesis, in vitro experiments and analyzed the data. M.T.G. performed cell-based experiments and analyzed the data. J.L.M. came up with the concept. J.L.M. and F.L. guided the research, and all the authors conceived experiments, interpreted the results, and participated in writing the manuscript.

## Additional information

**Competing interests:** The authors declare no competing interests.

