## [Peer Review File · Nature Communications]

Reviewers' comments:

Reviewer #1 (Remarks to the Author):

The authors report the study of a gold catalysed cyclisation in aqueous conditions and then in intracellular fashion. The manuscript then shows that gold and ruthenium based catalysis can be run concurrently in the same cells. This is potentially interesting as the ability to run different reactions in an orthogonal fashion within complex biological media may well become a potentially interesting area of future development. The study appears to have been performed carefully – I have a couple of specific queries that need to be addressed before a decision can be made – and contains the appropriate supporting information which is well presented. I am not an expert in the intricacies of performing and interpreting cellular studies (and hence potential pitfalls in experimental design or interpretation of data), but it seems to me that the appropriate control studies have generally been performed to establish suitable controls and confirm the results (e.g. checking cellular internalisation with Au).

This type of transformation has previously been shown in cells by Tanaka [Org Lett 2010 (ref 20)] using Au(III) catalysts and so the key discovery here is that LAuCl complexes are catalytically active in aqueous and biological media and do not need further activation by e.g. the addition of silver salts. Control studies nicely support the conclusion that the chloride dissociates in water relative to acetonitrile (though on a different complex to that ultimately used). This is then correlated to observed reactivity in cells. The gold complexes are added to the cells in a DMSO solution (which is a good ligand) which presumably affects this process, and seems worthy of consideration in the manuscript.

It is interesting that the authors show here how LAuCl complexes are more reactive than other LAuX systems in aqueous systems, and this could be of broader impact. Some of the general statements about LAuCl systems should be tempered though as they are not always strictly correct (as written): e.g. "However, it is well-known that the use of gold(I) chloride catalysts other than AuCl requires the addition of chloride scavengers, such as silver salts, to generate vacant sites in the gold." It is known that LAuCl complexes can be catalytically active in organic media (check many optimisation tables), but the chloride abstraction approach is widely used to generate (much) more reactive species, so for instance the following may be more accurate: "However, chloride scavengers such as silver salts are generally used to generate effective catalysts from gold(I) chloride complexes by aiding the generation of vacant coordination sites."

The ability to perform catalysis of small molecules in cells is of increasing interest. The authors show that the LAuCl complexes are active, and not poisoned by the complexity of the other intracellular material (e.g. supp Fig 4), enabling the desired transformation in a cell. Coupled to the insight into viable reactivity with LAuCl complexes, this is likely to lead to further interest in this field.

As a side-note this study (or others in the area) do not show whether catalysts affect any other molecules or processes in the cell (beyond looking at gross cellular toxicity on relatively short timeframes) – this seems increasingly important in terms of the longer term goals of these types of projects. While beyond the scope of this study, it would be possible to show whether any of the additives used in Supp Fig 4 were consumed.

The second major claim concerns orthogonal catalysis in cells with different metals. Figure 5 does show some red fluorescence in panel C (and presumably hence shows some reaction of 5 under gold catalysis) – How is this consistent with the statement that "Indeed, infrared fluorescence was only generated after incubation of Ru1 with substrate 5 (Fig. 5b, panels B, F and J)" or the claimed orthogonality (which is not the same as partial selectivity)?

Overall, while the manuscript could influence thinking in the field, the question over orthogonality seems critical especially given the precedence for this type of gold catalysed transformations in

cells.

Other corrections and queries:

"Interestingly, while the blue signal presents a more vesicular distribution, the green fluorescence is extended across the cytoplasm " This is interesting, how do the authors rationalize the different distribution of fluorescence from a single compound.

P2: 'Scientists' not 'scientist'

P2: 'this goal' not 'the goal'

Figure 1a – Could remove the picture of L-Au-Cl as this is already in the reaction scheme – the text can be integrated.

Figure 1a: 'Innocuous'

The statement "the unique ability of gold cationic complexes to activate π -bonds in a chemoselective manner," can be rephrased as it is not a unique property of gold - gold does show a very broad generality but other metals (e.g. platinum, but also Au(III) and other metals) or even non-metals can activate π -bonds in a chemoselective fashion.

Reviewer #2 (Remarks to the Author):

There are some interesting observations in this paper, but the treatment of some complicated gold chemistry is rather simplistic. Some of the statements about the chemistry are too general

Also, proof-of-principle studies in cells are fine, but how do the authors envisage a practical use for this technology? Formation of fluorescent molecules is one thing, but practical applications to other molecules is a much bigger problem. All they know is that some fluorescent product is generated in cells. Fluorescence itself is not quantitative and subject to many interferences (e.g. effect of environments of different polarity). There are no other data about the products of catalysis in cells apart from observation of fluorescence.

Major revision would be necessary before publication. Some points for attention are noted below

P2 (? Sorry – no page numbers on my pdf)

'Furthermore, gold complexes are redox stable,'

Certainly not true for many Au(III) complexes in aqueous solution, especially in the presence of thiols and thioethers. Au(0) sols are readily made from Au(III) chloride plus citrate.

'the chloride atom ensures'

the chloride ion ensures

'does not raise meaningful toxicities'

What is a meaningful toxicity?

'without cross contamination.'

Meaning?

'Gold(I) catalysis in aqueous media, and simultaneous intracellular Ru(II) and Au(I) catalysis'

There is no structure of a Ru complex in this figure.

Fig 2

Au5 is missing a + charge

'NMR studies revealed that the ^{31}P signal at 35.4 ppm corresponding to the gold chloride complex Au4 (in dry acetonitrile, using alendronic acid as internal standard in D₂O) shifts to 32.7 ppm when the complex is dissolved in water (see Supplementary Fig. 5),'

This is not a convincing experiment – see my comments below. More work is needed to investigate

the hydrolysis.

'Remarkably, when the reaction was carried out using the water insoluble AuClPPh₃ (Au₆), we observed no conversion (see Supplementary Table 2).'

Is this carried out in 1:4 v/v MeCN: water? Does no Au dissolve? If so not remarkable.

'This is particularly relevant when dealing with gold complexes, since it has been shown that some of them can be quite cytotoxic.⁴⁴

Ref 44 is a 2015 ref but work on the toxicity of gold phosphines dates back to Lorber et al in about 1979 and there have been many more reports since then.

'complex Au1 showed the lowest cytotoxicity among the eight complexes so far tested, with a cell viability of 80% at 25 μM after 6 h of incubation, while under the same conditions (25 μM), Au2 led to a decrease in the viability of 50%. Complexes with the carboxylate, Au3, and specially, sulfonate appendages, Au4, were considerably more toxic.'

IC50 values would be helpful for comparison of complexes.

'exhibiting good reaction rates in aqueous solvents'

What is a good reaction rate?

'undoubtedly due to the competitive coordination of the thiols to the gold.'

Need to be more specific: substitution for Au(I) likely reduction of Au(III)

'proteins like BSA (bovine albumin protein), which features many cysteines and histidines in their structure.'

BSA only has one free Cys (Cys34) and that is in a crevice and blocked in about 50% of commercial BSA.

'One equivalent of adenine, cytosine and histidine also had a poisoning effect,'

Why is this. Have interactions between the Au complexes and these molecules been checked, e.g. by NMR? I would expect binding to be very weak.

Since binding and equilibria are concentration-dependent, all these statements about interferences and mol equiv are not very meaningful unless concentrations are given.

'Finally, the transformation can also be carried out in presence of living bacteria'

Why is that relevant?

Table S1 is not very informative. What kind of bacteria and how many? Data are nowhere near sufficient to make such a sweeping statement

What kind of BSA was used? (fatty-acid-free? Or loaded? Purity?)

'Moreover, this complex (Au1') turned out to be more toxic than its corresponding chloride counterpart Au1'

Au1' might be converted into the chloride complex in the cell culture medium which probably contains about 0.1 M chloride.

'Gratifyingly, the gold complex Au1 showed a 50-fold increase of intracellular Au concentration with respect to non-treated blank cells,'

Surely they must contain more than the blank cells! How much is needed to justify gratification?!

'These results confirm that gold species can effectively enter living cells'

This has been known for a long time!

Figure 4

Error bars needed. Au concentration is high (75 μM)

What is the IC50 of Ru1?

Does Ru1 target mitochondria? If so, is that where the catalysis occurs?

SI

PS4

As drawn, complex Au5 is a Au(II) complex. Should have + charge.

PS5/3.2 and 3.4

What catalyst is used in this reaction. It says on the arrow [AuPPh3] is the metal complex- impossible.

If the NTf is coordinated it should be inside the Au bracket. Is there an x-ray structure of this complex?

PS13/S14

Peak assignments need to be added

X axis needs a label

PS20

Do they see intermediate spectra with peaks for Cl and H2O adducts?

What does 5 min mean? How long did it take to acquire the spectra?

Because of susceptibility effects, the reference may not have the same resonance frequency in both solvents.

Why not add H2O to the MeCN sample to monitor hydrolysis?

Fig S6: 'MS spectra of the reaction using gold complex Au4 and piperidine'

What does this mean? 3-coordinate Au(I) complexes are known but unusual.

PS22

'before treatment with different concentrations of the gold complexes (25 μ M and 50 μ M)'

These are high concentrations. What are the IC50 values?

Figure S7

Error bars are needed

S10

were incubated with either catalyst Au1-Au8 (75 μ M)

This is surely quite a toxic dose.

PS26

Where is the synthesis and characterization of Ru1 given?

'Cells were incubated with Ru1 (25 μ M) or Au1 (50 μ M) for 30 min,'

How much Ru and Au is taken up in this time? How many cells die in the 6 hours of the experiment?

We would like to thank the referees for their constructive and positive comments. Following their recommendations, we have carefully revised the manuscript to appropriately address all their concerns and suggestions. As a result, we have assembled a more robust and sound article. Changes have been highlighted in the revised manuscript, and in a pdf copy of the supplementary information.

In addition to other novelties (catalysis by gold chloride in aqueous media and cells, activation by water, novel allyl caged probes for the ruthenium catalysis, etc), our paper reports the first examples of simultaneous, orthogonal and biorthogonal metal-catalyzed reactions occurring inside living cells. As suggested by referee 1, this may open a new and interesting area for future development.

Detailed point by point reply to the reviewers:

Reviewer 1:

This type of transformation has previously been shown in cells by Tanaka [Org Lett 2010 (ref 20)] using Au(III) catalysts and so the key discovery here is that LAuCl complexes are catalytically active in aqueous and biological media and do not need further activation by e.g. the addition of silver salts.

Response: This previous work by Kim and Kim (*Org. Lett*, **2010**, 12, 932) consists on a fluorescent sensing strategy for detection of Au³⁺ ions (using HAuCl₄ as analyte), therefore quite different from our AuCl(L)-mediated catalytic strategies. Furthermore, the proposed reaction has been carried out in **fixed** cells, not living cells. Anyhow, the reference has been cited.

The gold complexes are added to the cells in a DMSO solution (which is a good ligand) which presumably affects this process, and seems worthy of consideration in the manuscript

Response: We have now run control experiments under optimized conditions in the presence of different excess amounts of DMSO, and we did not observe any effect in the reaction outcome. This control experiment has been added to the supplementary information (section 6, page S23).

Some of the general statements about LAuCl systems should be tempered though as they are not always strictly correct (as written): e.g. "However, it is well-known that the use of gold(I) chloride catalysts other than AuCl requires the addition of chloride scavengers, such as silver salts, to generate vacant sites in the gold." It is known that LAuCl complexes can be catalytically active in organic media (check many optimisation tables), but the chloride abstraction approach is widely used to generate (much) more reactive species, so for instance the following may be more accurate: "However, chloride scavengers such as silver salts are generally used to generate effective catalysts from gold(I) chloride complexes by aiding the generation of vacant coordination sites."

Response: We do agree with the referee. The sentence was accordingly revised in the manuscript. However, it is worth to mention that in our case, the AuCl(L) complex **Au1**, in MeCN, and without silver salts, is not catalytically active (entry 6, Table 1).

As a side-note this study (or others in the area) do not show whether catalysts affect any other molecules or processes in the cell (beyond looking at gross cellular toxicity on relatively short timeframes) – this seems increasingly important in terms of the longer term goals of these types of projects. While beyond the scope of this study, it would be possible to show whether any of the additives used in Supp Fig 4 were consumed.

Response: Yes, the referee is right, the field is yet in its infancy, and much remains to be learned on the detailed cellular consequences of these processes. While the cellular analysis is not possible, we have carried out the reaction under the optimized aqueous conditions in the presence of 1 equivalent of tyrosine as additive. After 24 h, the crude residue was analyzed by LC/MS-ESI and NMR, which confirmed that tyrosine remains intact. The information is included in the supplementary information (section 6.2, page S21).

The second major claim concerns orthogonal catalysis in cells with different metals. Figure 5 does show some red fluorescence in panel C (and presumably hence shows some reaction of **5** under gold catalysis)

Response: This fluorescence in panel C is residual, and highly probably coming from the substrate. Indeed, panel A shows similar residual levels of red fluorescence in some cells. If we had a minimally significant activity of the gold catalyst we should observe fluorescence levels like in panel D. Cross-reactivity experiments included in figure S13 were also fully consistent with these observations. Anyway, the lack of reactivity was confirmed by in vitro experiments (Supplementary information, page S33). Therefore, treatment of precursor **5** with one equivalent of gold catalyst **Au1** for 15 h in H₂O : MeCN (8:2) didn't produce even traces of the deprotected product.

Interestingly, while the blue signal presents a more vesicular distribution, the green fluorescence is extended across the cytoplasm” This is interesting, how do the authors rationalize the different distribution of fluorescence from a single compound

Response: It is well known that this type of tricyclic amines can present different fluorescence depending on the polarity and environment (*Chem. Sci.*, **2017**, 8, 1915).

P2: 'Scientists' not 'scientist'

Response: fixed.

P2: 'this goal' not 'the goal'

Response: fixed.

Figure 1a – Could remove the picture of L-Au-Cl as this is already in the reaction scheme – the text can be integrated.

Response: Figure 1a has been revised according to this suggestion.

Figure 1a: 'Innocuous'

Response: fixed.

The statement “the unique ability of gold cationic complexes to activate π -bonds in a chemoselective manner,” can be rephrased as it is not a unique property of gold - gold does show a very broad generality but other metals (e.g. platinum, but also Au(III) and other metals) or even non-metals can activate pi-bonds in a chemoselective

fashion.

Response: We have removed the word “unique”.

Reviewer 2

Also, proof-of-principle studies in cells are fine, but how do the authors envisage a practical use for this technology? Formation of fluorescent molecules is one thing, but practical applications to other molecules is a much bigger problem. All they know is that some fluorescent product is generated in cells. Fluorescence itself is not quantitative and subject to many interferences (e.g. effect of environments of different polarity). There are no other data about the products of catalysis in cells apart from observation of fluorescence.

Response: The area of metal catalysis in biological settings is still in its infancy; however, previous work carried out by Bradley, Zimmerman and others has already shown the viability of using this technology for the in situ generation of drugs. Moreover, associating the catalysts to specific targets would allow an amplified and site selective generation of active compounds. This is a very exciting prospect for the potential application of this technology.

The work described in our article is yet in a conceptual, proof of principle stage; and the best current way of monitoring the intracellular activity of our new catalytic strategies is based on fluorescence microscopy, which is also a standard technique in cell biology. Quantifying catalytic activities and turnover rates in living settings is extremely challenging, but should be approached in the near future.

“Furthermore, gold complexes are redox stable” Certainly not true for many Au(III) complexes in aqueous solution, especially in the presence of thiols and thioethers. Au(0) sols are readily made from Au(III) chloride plus citrate

Response: We refer to gold(I) complexes. We have revised the text to ““Furthermore, gold(I) complexes are fairly redox stable”.

‘the chloride atom ensures’

Response: “Atom” changed to “ion”.

‘does not raise meaningful toxicities’

What is a meaningful toxicity?

Response: Changed to: “significant”. With 25 μM of the gold complex, we observed 100% viability after 6 h, and with 50 μM , the viability is higher than 80%.

‘without cross contamination.’

Meaning?

Response: Changed to “cross-reactivity”.

‘Gold(I) catalysis in aqueous media, and simultaneous intracellular Ru(II) and Au(I) catalysis’ There is no structure of a Ru complex in this figure.

Response: Figure 1 is introductory, and intends to illustrate general concepts; therefore, we modestly consider that the exact structures of the **Au** and **Ru** complexes are better introduced later in the discussion.

Fig 2, Au5 is missing a + charge

Response: fixed.

Remarkably, when the reaction was carried out using the water insoluble AuClPPh₃ (Au6), we observed no conversion (see Supplementary Table 2).’ Is this carried out in 1:4 v/v MeCN: water? Does no Au dissolve? If so not remarkable

Response: The reaction was carried out in H₂O:MeCN (8:2) like the other reactions, and we blame the lack of activity to the low solubility of this complex in the mixture. We have changed the sentence to “when the reaction was carried out using AuCl(PPh₃) (**Au6**), we observed no conversion, most probably because of the low solubility of the complex”.

‘This is particularly relevant when dealing with gold complexes, since it has been shown that some of them can be quite cytotoxic.44’

Ref 44 is a 2015 ref but work on the toxicity of gold phosphines dates back to Lorber et al in about 1979 and there have been many more reports since then.

Response: Reference 44 (*Chem. Soc. Rev.* **2015**, *44*, 8786) is a very detailed review covering most of the gold(I) and gold(III) toxic complexes described and is the latest published in this field so we believed that this reference is enough to cover the field of toxic gold complexes. Nevertheless, we have included the reference suggested by the referee 2 as reference 45.

IC50 values would be helpful for comparison of complexes.

Response: IC50 values have been calculated and included in the supplementary figure S10 (c), page S29.

‘exhibiting good reaction rates in aqueous solvents’

What is a good reaction rate?

Response: This sentence was changed to: “In addition to exhibiting good reaction rates in aqueous solvents (high yields in less than 2-3 h)”.

‘undoubtedly due to the competitive coordination of the thiols to the gold.’ Need to be more specific: substitution for Au(I) likely reduction of Au(III)

Response: We are using Au(I) (no Au(III)) complexes, therefore the inhibition must be due to coordination.

‘proteins like BSA (bovine albumin protein), which features many cysteines and histidines in their structure.’ BSA only has one free Cys (Cys34) and that is in a crevice and blocked in about 50% of commercial BSA.

Response: The referee is right. The BSA protein has 35 Cys, but 34 of them are as disulfide bridges, and only one with the free thiol. We have accordingly revised the text to “which features one free cysteine and several histidines in their structure”.

‘One equivalent of adenine, cytosine and histidine also had a poisoning effect,’

Why is this. Have interactions between the Au complexes and these molecules been checked, e.g. by NMR? I would expect binding to be very weak.

Response: As requested, we have specifically analyzed the interaction between some of these additives, (adenine), and the gold complexes. The results, included in the supplementary information in section S8, page S27, confirm that the adenine can coordinate the gold atom.

Since binding and equilibria are concentration-dependent, all these statements about interferences and mol equiv are not very meaningful unless concentrations are given.

Response: The concentrations are given in the supplementary information page S21: "Substrate **3** (0.050 mmol, 15.90 mg) and glycine (0.050 mmol, 3.70 mg, 1.0 eq.) were added to a Schlenk tube containing a stir bar followed by the addition of [AuCl(PTA)] (**Au1**, 0.005 mmol, 1.90 mg, 10.0 mol%). MeCN (200 μ L) was added and the reaction mixture was stirred at 300 rpm until a homogenous solution was formed (10 s). H₂O (800 μ L) was added (final volume 1.0 mL, [Substrate] = 50.0 mM), the Themowatch-controlled heating block was fixed at 37 °C and the reaction was stirred for 24 h".

'Finally, the transformation can also be carried out in presence of living bacteria'

Why is that relevant?

Response: In addition to further show the biocompatibility of our catalytic systems, being able to achieve amplifying catalytic transformations in bacteria might provide the basis for the future development of new pharmacological treatments or diagnosis techniques. Indeed, other groups (Tirrell, Chen, Ward...) have paid a lot of attention to the development of Click or metathesis reactions in bacteria.

What kind of bacteria and how many? Data are nowhere near sufficient to make such a sweeping statement

What kind of BSA was used? (fatty-acid-free? Or loaded? Purity?)

Response: This information has now been detailed in the supplementary information (section S1, page S3).

'Moreover, this complex (Au1') turned out to be more toxic than its corresponding chloride counterpart Au1' Au1' might be converted into the chloride complex in the cell culture medium which probably contains about 0.1 M chloride

Response: The reviewer is right. The culture cell media used in the studies contains 0.12 M chloride. However, we didn't observe reactivity (fluorescence) when cells were incubated with the triflimide **Au1'**, suggesting that it is not converted to the chloride **Au1**.

'Gratifyingly, the gold complex Au1 showed a 50-fold increase of intracellular Au concentration with respect to non-treated blank cells,'

Surely they must contain more than the blank cells! How much is needed to justify gratification?!

Response: We have removed "gratifyingly".

'These results confirm that gold species can effectively enter living cells'

This has been known for a long time!

Response: The referee is right, but in most of these reports, carried out in the context of toxicity studies, there were not compared results on the relative internalization of different species.

Figure 4

Error bars needed. Au concentration is high (75 μ M)

Response: Error bars have been now included in Figure 4c. While we observed good intracellular reactivity with concentrations of 50 μ M (and even 25 μ M) of the best gold complexes, we preferred to do the analysis of cell uptake and toxicity with higher concentrations, in order to facilitate the comparison among different catalysts.

What is the IC₅₀ of Ru1?

Does Ru1 target mitochondria? If so, is that where the catalysis occurs?

Response: The ruthenium complex **Ru1** has been previously described by our group in *Nat. Commun.* **7**, 12538-12547 (2016). This complex displays a phosphonium moiety which is known to be a directing vector to the mitochondria. Indeed, **Ru1** partially targets this organelle. The catalysis takes place both in the mitochondria and across the cytoplasm. Toxicity studies of ruthenium complexes are displayed in the mentioned paper. We could observe a decrease in the viability of cells (of around 30%) only at concentrations above 100 μM , and after 24 h. For concentrations below 50 μM no decrease in viability was observed after 24 h.

PS4

As drawn, complex Au5 is a Au(II) complex. Should have + charge.

Response: The referee 2 is right and we have corrected the structure of the complex **Au5** and added the counterion (SbF_6).

PS5/3.2 and 3.4

What catalyst is used in this reaction. It says on the arrow $[\text{AuPPh}_3]$ is the metal complex- impossible.

If the NTf is coordinated it should be inside the Au bracket. Is there an x-ray structure of this complex?

Response: The gold complex used for the synthesis of some of the substrates of this work is commercial available from *Sigma Aldrich* as [Bis(trifluoromethanesulfonyl)imidate](triphenylphosphine)gold(I) (2:1) toluene adduct with SKU 677922.

PS13/S14

Peak assignments need to be added

X axis needs a label

Response: According to the suggestion of the referee, the peak assignments have been added with the label of the X axis (ppm).

PS20

Do they see intermediate spectra with peaks for Cl and H₂O adducts? What does 5 min mean? How long did it take to acquire the spectra?

Response: As shown in the Figure S7, we only observed two peaks in ^{31}P NMR (one for the internal standard on the other one for the Cl or H₂O adducts) and not any intermediate or other signal (the width of the spectra is around 400 ppm).

"5 min" means that spectra were recorded after 5 min.

The number of the scans were the same for both ^{31}P NMR spectra (48 scans with a relaxing delay of 2 s).

Following the reviewer request, we have included the protocol for obtaining these ^{31}P NMR spectra (section S7, page S24).

Because of susceptibility effects, the reference may not have the same resonance frequency in both solvents.

Response: We have included the detailed protocol for the ^{31}P NMR experiments in the supplementary information (section S7, page S24). The susceptibility effects can be discarded in the experiment because the internal standard (alendronic acid) was dissolved in a capillary tube in D₂O, and this capillary tube was introduced into an NMR tube in which the Au complex was dissolved in the corresponding solvent.

Why not add H₂O to the MeCN sample to monitor hydrolysis?

Response: We have considered this experiment, but MeCN is a known gold coordinating agent (there are a variety of gold complexes using benzonitrile or acetonitrile as ligands), and thus we discarded it.

Fig S6: 'MS spectra of the reaction using gold complex Au4 and piperidine'

What does this mean? 3-coordinate Au(I) complexes are known but unusual

Response: The reviewer is right. The assignment to the *m/z* peak at 709.51 corresponds to complex with a second piperidine moiety, however it was drawn in a wrong way. The second piperidine must be a counterion to the sulfonate pendant. We have accordingly modified the figure S8 in the supplementary information.

PS22

'before treatment with different concentrations of the gold complexes (25 μ M and 50 μ M)'

These are high concentrations. What are the IC50 values?

Response: These are usual metal complex concentrations for this type of studies. IC50 values have been calculated and are now shown in figure S10 of the supplementary information (section S10, page S29).

Figure S7

Error bars are needed

Response: Error bars have been included in Figure S10.

S10

were incubated with either catalyst Au1-Au8 (75 μ M)

This is surely quite a toxic dose.

Response: Even with these relatively high concentrations, the cell viability after 6 hours is higher than 70% in all cases, except for **Au3**, **Au4** and **Au1'** (between 60 and 70%). The differences of toxicity depending on the ligand suggest that there is space for further tuning the activity/toxicity relationship in future work.

PS26

Where is the synthesis and characterization of Ru1 given?

Response: The ruthenium complex **Ru1** has been previously reported by the group in *Nat. Commun.* **7**, 12538-12547 (2016). The synthesis and characterization can be found in the Supplementary Information of this publication.

'Cells were incubated with Ru1 (25 μ M) or Au1 (50 μ M) for 30 min,'

How much Ru and Au is taken up in this time? How many cells die in the 6 hours of the experiment?

Response: Toxicity studies of ruthenium complexes are displayed in the mentioned paper (*Nat. Commun.* **7**, 12538-12547, 2016). We could observe a decrease in the viability of cells (of around 30%) only at concentrations above 100 μ M and after 24 h. For concentrations below 50 μ M no decrease in viability was observed after 24 h. Since the complex is not toxic under these conditions, no IC50 can be calculated at 25 μ M. Toxicity studies carried out for **Au1** complex have already been shown in this work (Fig. S10), with a cell survival of approx. 80% after 6 h of incubation at 50 μ M.

ICP studies for the internalization of these complexes under these specific conditions have been performed and the results are shown in Supplementary Figure S13, page S32:

Ru1 (25 μ M) = 117.5 ng/10⁶ cells

Au1 (50 μ M) = 148.8 ng/10⁶ cells.

Overall, we have worked to respond all the reviewers concerns and requests. We thank them for their effort to make constructive criticisms. We are very sure that this multidisciplinary article will call the interest of a varied audience including chemists, biologists and medicinal researchers, and represents a excellent contribution to Nature Comm..

Reviewers' comments:

Reviewer #2 (Remarks to the Author):

Response: We refer to gold(I) complexes. We have revised the text to "Furthermore, gold(I) complexes are fairly redox stable".

But only if there is a strong pi-acceptor ligand bound to Au(I).

There is a need for clarity because this paper certainly does deal with both Au(I) and Au(III) complexes. For example there are IC50 values in Fig S10 for Au(III) complexes 5, 7 and 8. These are unstable with respect to both ligand substitution and reduction in cell culture media.

Response: The reaction was carried out in H₂O:MeCN (8:2) like the other reactions, and we blame the lack of activity to the low solubility of this complex in the mixture. We have changed the sentence to "when the reaction was carried out using AuCl(PPh₃) (Au6), we observed no conversion, most probably because of the low solubility of the complex".

This is unsatisfactory and cannot be left like this. The solubility could be determined. Another mixture of solvents could be tried so as to increase solubility.

Response: IC50 values have been calculated and included in the supplementary figure S10 (c), page S29.

Error bars need to be added to the figures and table. How many experiments? 3 replicates?

Response: As requested, we have specifically analyzed the interaction between some of these additives, (adenine), and the gold complexes. The results, included in the supplementary information in section S8, page S27, confirm that the adenine can coordinate the gold atom. On the mass spectrum, the structure of M is shown as a +1 ion, so M+H and M+Na would be 2+ ions

Response: The reviewer is right. The culture cell media used in the studies contains 0.12 M chloride. However, we didn't observe reactivity (fluorescence) when cells were incubated with the triflimide Au1', suggesting that it is not converted to the chloride Au1.

This is hard to believe. Triflimide is surely a very weak donor and should be displaced by chloride. Experiments are needed to compare NMR and MS spectra in presence and absence of 0.1 M chloride.

Response: Error bars have been now included in Figure 4c. While we observed good intracellular reactivity with concentrations of 50 μM (and even 25 μM) of the best gold complexes, we preferred to do the analysis of cell uptake and toxicity with higher concentrations, in order to facilitate the comparison among different catalysts.

Need to say what the error bars represent (SEs?). They are big so checks are needed as to which are statistically different e.g. using a t-test

Page 5

"NMR studies revealed that the ³¹P signal at 35.4 ppm corresponding to the gold chloride complex Au4 (in dry acetonitrile, using alendronic acid as internal standard in D₂O) shifts to 32.7 ppm when the complex is dissolved in water (see Supplementary Fig. S7), change consistent with the formation of the cationic gold(I)-aquo derivative"

More work is needed to allow interpretation of the NMR data – need to add chloride and monitor the shift in water for example.

And to my question

"Why not add H₂O to the MeCN sample to monitor hydrolysis?"

they responded:

Response: We have considered this experiment, but MeCN is a known gold coordinating agent (there are a variety of gold complexes using benzonitrile or acetonitrile as ligands), and thus we discarded it.

So if that is the case the NMR spectrum may be that of the MeCN complex and not the chlorido complex?

Page 11

"bearing a triflimide (NTf₂) instead of a Cl counterion ([Au(PTA)]NTf₂, Au¹⁺),"

A confusing formula with one-coordinate Au(I) and uncoordinated Ntf.

NTf is not defined in the script. Is it neutral or negatively charged?

In Si section 4.6 I find the use of AgNtf₂ so I assume NTf is an anion (deprotonated NH?), in which case this is a Ag(II) complex – very unusual, strong oxidant I assume. The Au complex would also be Au(II) as written.

The formula is written with one-coordinate Au(II) and two non-coordinated NTfs

Very confusing.

These experiments must be reported in a way in which they can be repeated. As far as I can see there is no mention of the source of the NTf compounds.

I asked previously:

"What catalyst is used in this reaction. It says on the arrow [AuPPh₃] is the metal complex- impossible. If the NTf is coordinated it should be inside the Au bracket. Is there an x-ray structure of this complex?"

The response below is unacceptable.

Response: The gold complex used for the synthesis of some of the substrates of this work is commercial available from Sigma Aldrich as

[Bis(trifluoromethanesulfonyl)imidate](triphenylphosphine)gold(I) (2:1) toluene adduct with SKU 677922.

I cannot find SKU 677922. Is a Google search.

Page 12

These results confirm that gold species can effectively enter living cells, but also point out that the entrance depends on the characteristics of the ligand,

And on the redox behaviour. The Au(II) complexes are likely to be reduced in the culture medium. Characteristics (spelling)

Response: The reviewer is right. The assignment to the m/z peak at 709.51 corresponds to complex with a second piperidine moiety, however it was drawn in a wrong way. The second piperidine must be a counterion

As drawn in Fig S8, M is a +1 ion so M+H would carry a +2 charge

Reviewer #3 (Remarks to the Author):

In this manuscript Mascareñas et al. describe a bioorthogonal gold-catalyzed C-C bond-forming transformation which can be carried out in live cells in parallel to the previously reported by the same group Ru(II)-catalyzed decaging reaction. The fast developing field of organometallic chemical biology provides many opportunities for control over biomolecule structure and function as well as small molecule manipulation in cellular environments. In this context, gold catalysis has

been historically lagging behind due to the potential for strong catalyst inactivating Au-S bonding. It is therefore exciting to see more recent reports by Tanaka et al., Unciti-Broceta et al. [ACIE 2017, 129, 12722] and Mascareñas et al. introducing new gold-catalyzed transformations to the repertoire of available reactions for the chemical biology settings. The authors of this manuscript have addressed the previous comments, and I believe that the findings of this work are significant enough to be published as an article in Nature Communications after several minor changes are implemented as listed below.

P3: "ease access" needs to be changed to "easy access";

P3: "catalytic network" should be removed since the two processes are not connected and therefore are not part of a "network" (in fact, orthogonal as described in the paper);

P4: Figure 1 should be changed. "first gold-catalyzed C-C bond forming reaction in water" needs to be modified as technically "gold-catalyzed C-C bond forming reactions in water" have been previously reported (e.g. see Wei, C., Li, C.-J. JACS, 2003, 125, 9584); This statement should also be modified at the end of the manuscript. Furthermore, "complete chemo- and regioselectivity" should be changed to "complete chemoselectivity" as the substrates were carefully chosen to achieve regioselectivity of the transformations.

P5: The authors might consider removing figure 2, as it doesn't bear any important information and only confuses the reader. On the contrary, comparison of cytotoxicities of tested gold complexes is an important piece of data, which should be presented in the main text of the paper (e.g. MTT assay), perhaps later in the text.

P5: It would be good to see the comparison of ³¹P chemical shifts in acetonitrile and water for the main complex (Au1) used in the study.

P6: Taking into account the inhibitory effect of cysteine and glutathione in in vitro studies it will be interesting to hear the authors' thoughts/rationale for the efficiency of the catalytic process in cells, where multiple reactive cysteines and glutathione are present, which must deactivate the catalyst.

POINT BY POINT reply to the Reviewers Comments. The requests and replies of the first revision are also included in blue.

Comments and answers to Reviewer 2

1. Regarding our initial sentence: "Furthermore, gold complexes are redox stable"

Reviewer (1st revision): *Certainly not true for many Au(III) complexes in aqueous solution, especially in the presence of thiols and thioethers. Au(0) sols are readily made from Au(III) chloride plus citrate*

Reply: We refer to gold(I) complexes. We have revised the text to "Furthermore, gold(I) complexes are fairly redox stable".

Reviewer (2nd revision): *But only if there is a strong pi-acceptor ligand bound to Au(I).*

There is a need for clarity because this paper certainly does deal with both Au(I) and Au(III) complexes. For example there are IC50 values in Fig S10 for Au(III) complexes 5, 7 and 8. These are unstable with respect to both ligand substitution and reduction in cell culture media.

Reply: We wanted to remark that gold(I) complexes tend to be stable to air, and many of them have also been described to be rather stable to reduction, particularly when compared to gold(III), see for instance: A. Cassini *Bioorganometallic Chemistry: Applications in Drug Discovery, Biocatalysis, and Imaging*, First Edition (2015). However, we admit that the sentence can be misleading, and therefore we changed it to: "Furthermore, the reactions promoted by gold complexes, especially by gold(I) species, tend to be tolerant to air and moisture".

2. Regarding our initial sentence: "Remarkably, when the reaction was carried out using the water insoluble AuCIPPh₃ (**Au6**), we observed no conversion (see Supplementary Table 2)."

Reviewer (1st revision): *Is this carried out in 1:4 v/v MeCN: water? Does no Au dissolve? If so not remarkable*

Reply: The reaction was carried out in H₂O:MeCN (8:2), like the other reactions, and we blame the lack of activity to the low solubility of this complex in the mixture. We have changed the sentence to "when the reaction was carried out using AuCl(PPh₃) (**Au6**), we observed no conversion, most probably because of the low solubility of the complex".

Reviewer (2nd revision): *This is unsatisfactory and cannot be left like this. The solubility could be determined. Another mixture of solvents could be tried so as to increase solubility.*

Reply: Since the result was somewhat secondary in the context of the paper, we didn't investigate the origin of the lack of reactivity, but we hypothesized that it could be due to the low water solubility of **Au6**, that precludes its activation. We apologize for not addressing the point with further detail in the first revision.

In this second revision, to fulfill the reviewer's requests, we present a more detailed study included in the Supplementary Information (Section S10). In particular, to compare and assess the solubility of complexes **Au1** and **Au6**, we have analyzed by ³¹P-NMR several solutions of equimolar mixtures of **Au1** and **Au6** in different deuterated solvents. When DMSO-d₆ was used as solvent, a 1:1 ratio between the two phosphorus signals corresponding to both complexes was observed, suggesting that, from a qualitative point of view, **Au1** and **Au6** are equally soluble in this solvent. However, in MeCN-d₃ the integrals showed a **Au1:Au6** ratio of 1:2, confirming that **Au6** is significantly more soluble than **Au1** in this solvent. Finally, in the deuterated mixture of solvents employed for the catalytic reactions (D₂O:MeCN-d₃ = 8:2) only the signal of **Au1** was detected, confirming that **Au6** is not soluble in this mixture.

This result is in line with our initial assumption that the poor reactivity of **Au6** is due to its lack of solubility in a milieu with a high proportion of water, which hinders its water-promoted activation. On the other hand, and consonance with the requirement of water to activate the gold chloride, complex **Au6** does not catalyze the conversion of **3** into **4** in acetonitrile, unless a chloride scavenger (such as AgSbF_6) is added.

3. **Reviewer (1st revision):** *IC50 values would be helpful for comparison of complexes.*

Reply: IC50 values have been calculated and included in the supplementary figure S10 (c), page S29.

Reviewer (2nd revision): *Error bars need to be added to the figures and table. How many experiments? 3 replicates*

Reply: In Supplementary Fig. S17a, error bars had already been added. In Supplementary Fig. S17b we didn't include the error bars because the error in these measurements was less than 6% (this explanation is written in the caption). We have now included the error bars in the table, as requested. The error bars represent the standard deviation of three different samples (comment included in the footnote).

4. Regarding our sentence: "One equivalent of adenine, cytosine and histidine also had a poisoning effect"

Reviewer (1st revision): *Why is this. Have interactions between the Au complexes and these molecules been checked, e.g. by NMR? I would expect binding to be very weak.*

Reply: As requested, we have specifically analyzed the interaction between some of these additives, (adenine), and the gold complexes. The results, included in the Supplementary Information, Section S6, confirm that the adenine can coordinate the gold atom.

Reviewer 2 (2nd revision): *On the mass spectrum, the structure of M is shown as a +1 ion, so M+H and M+Na would be 2+ ions.*

Reply: In fact, the labels that we used for the assignment were not accurate. We apologize for the inconvenience. To avoid any misinterpretation, the Supplementary Fig. S3 was now modified to include the molecular structure of the two adenine-gold complexes detected by ESI-MS ($m/z = 674$ and 696). Both of them are +1 ions (bearing either a sulfinic acid moiety or a sodium sulfinic salt).

5. Regarding our sentence: 'Moreover, this complex (**Au1'**) turned out to be more toxic than its corresponding chloride counterpart **Au1**

Reviewer (1st revision): ***Au1'** might be converted into the chloride complex in the cell culture medium which probably contains about 0.1 M chloride*

Reply: The reviewer is right. The culture cell media used in the studies contains 0.12 M chloride. However, we didn't observe reactivity (fluorescence) when cells were incubated with the triflimide **Au1'**, suggesting that it is not converted to the chloride **Au1**.

Reviewer (2nd revision): *This is hard to believe. Triflimide is surely a very weak donor and should be displaced by chloride. Experiments are needed to compare NMR and MS spectra in presence and absence of 0.1 M chloride.*

Reply: First of all, it is important to note that while "[NTf₂]" should be formally named as bis(trifluoromethanesulfonyl)imidate or bistriflimidate, the common term "triflimide" can be also employed. The referee is right, bistriflimidate ([NTf₂]⁻) is a weak donor, and a labile ligand. Indeed, this has complicated the analysis of its reactivity by ESI-MS, owing to its low stability under the ionization conditions, and the tendency to form higher order species. In order to learn more on the performance of this gold bistriflimidate (or triflimide)

we made larger amounts using an improved synthetic protocol (see Supplementary Information, section S2.6), and repeated reactivity experiments both in vitro and in cells. Cell experiments confirmed the lower toxicity and much better performance of the chloride **Au1** over the bistriflimidate **Au1'** (see Fig. S20). Importantly, parallel in vitro experiments with chloride **Au1** and **Au1'**, carried out in cell culture medium DMEM:MeCN 8:2, confirmed a good reactivity with the chloride (94%, Table S3, entry 2), and almost no reaction with the bistriflimidate (Table S3, entry 6).

These data suggest that the gold bistriflimidate Au1' species is rapidly deactivated in biologically complex media (containing thiols, proteins, etc).

The studies proposed by the reviewer to prove if chloride can displace the NTf₂ group were carried out with the bistriflimidate version of the triarylphosphine complex, **Au4'**, owing to its better solubility in water, which facilitates the analysis, and in consonance with other studies carried out with **Au4** (see point 7 below). ESI-MS data indicated that after addition of NaCl to a D₂O solution of **Au4**, a mixture of species including the gold(I) chloride complex **Au4** is detected (see section S9). **These data confirm that chloride can displace the bistriflimidate, as suggested by the reviewer.**

However, in complex media like DMEM, there are many potential nucleophiles that can compete, and somewhat deactivate this gold-bistriflimidate complex.

6. *Reviewer: "Error bars needed. Au concentration is high (75 uM)"*

Reply: Error bars have been now included in Fig. 4c. While we observed good intracellular reactivity with concentrations of 50 μ M (and even 25 μ M) of the best gold complexes, we preferred to do the analysis of cell uptake and toxicity with higher concentrations, in order to facilitate the comparison among different catalysts.

Reviewer (2nd revision): *Need to say what the error bars represent (SEs?). They are big so checks are need as to which are statistically different e.g. using a t-test*

Reply: The error bars in Fig. 4c represent the standard deviation of three different measurements of the same sample, since they are ICP measurements. The referee is right and the values were large due to a technical problem with the processing program. The values represented in the previous graphic didn't correspond to the real deviation standard values, since the program introduced fixed and equal error bars for each bar. We apologize for overlooking this error. Now, this has been revised, so the real values are represented in the graphic, with an error < 6%.

7. *Regarding the paragraph of page 5: "NMR studies revealed that the ³¹P signal at 35.4 ppm corresponding to the gold chloride complex Au4 (in dry acetonitrile, using alendronic acid as internal standard in D₂O) shifts to 32.7 ppm when the complex is dissolved in water (Supplementary Fig. S7), change consistent with the formation of the cationic gold(I)-aquo derivative"*

Reviewer (2nd revision): *More work is needed to allow interpretation of the NMR data – need to add chloride and monitor the shift in water for example*

Reply: To complete the studies with these complexes we have now carried out new NMR and MS experiments which are included in the Supplementary Information (Section S8).

Complex **Au4** is characterized by a singlet signal at 32.96 ppm in the ³¹P-NMR spectrum in water. When this solution was analyzed by ESI-MS, the aquo-gold complex was observed, together with the original gold(I)-chloride species (Supplementary Fig. S8). These results confirm that gold(I)-aquo species are partially

generated from the gold(I)-chloride complexes in water. Accordingly, we propose that the ^{31}P -NMR signal at 32.96 ppm represents the average of different gold(I)-chloride and gold(I)-aquo species that are in equilibrium. On the other hand, when **Au4** is dissolved in MeCN (^{31}P -NMR signal at 35.35 ppm), the ESI-MS only shows the gold chloride complex ($m/z = 618.96$), confirming that, in the absence of water, the Au-Cl bond is not ionized (Supplementary Fig. S11).

As suggested by the Reviewer, we have also monitored the addition of chloride ions to the solution of **Au4** in water (Supplementary Fig. S9). As expected, we have effectively observed a shift in the phosphorus signal from 32.96 ppm, in pure water, to 33.72 ppm, in a saturated solution of NaCl. Analysis of the latter solution by ESI-MS revealed the disappearance of the signals corresponding to the gold(I)-aquo complex, indicating that the equilibrium has been shifted towards the gold(I)-chloride species [**Au4**].

Moreover, to further exclude that the change in ionic strength of the solvent was behind the shift in ^{31}P -NMR, we have measured the spectrum of the phosphine ligand of complex **Au4** in pure water and in a saturated solution of NaCl, observing in both cases the same signal at -4.96 ppm (Supplementary Fig. S10).

After these results, we have modified the main text in the manuscript (highlight in yellow in page 5), to better describe these new observations.

8. **Reviewer (1st revision)** Why not add H₂O to the MeCN sample to monitor hydrolysis?

Reply: We have considered this experiment, but MeCN is a known gold coordinating agent (there are a variety of gold complexes using benzonitrile or acetonitrile as ligands), and thus we discarded it.

Reviewer (2nd revision): So if that is the case the NMR spectrum may be that of the MeCN complex and not the chlorido complex?

Reply: We are not sure which particular NMR spectrum the Reviewer is referring to. Nevertheless, we provide now a complete explanation of the process, supported by NMR and ESI-MS experiments:

It is well known that acetonitrile cannot break by itself the Au-Cl bond in this type of phosphine gold(I) complexes [AuCl(L)]. Indeed, the ESI-MS of an acetonitrile solution of complex **Au4** only shows the [AuCl(L)] species (Supplementary Fig. 8c).

However, if the Au-Cl bond is broken, for instance by addition of a silver salt, and acetonitrile is present in the solution, the formation of [Au(NCMe)(L)]⁺ species might be expected (*NOTE: indeed, the way to prepare such complexes consists of mixing an acetonitrile solution of [AuCl(L)] species with AgX to yield [Au(NCMe)(L)]⁺X⁻ species and AgCl, which precipitates*).

^{31}P -NMR monitoring of a solution of **Au4** in MeCN, upon addition of increasing amounts of H₂O, revealed a progressive upper field shift of the initial signal (35.35 ppm, Supplementary Fig. S12). ESI-MS analysis of these aquo/MeCN mixtures allowed to observe peaks corresponding to the gold(I)-chloride, gold(I)-aquo and gold(I)-acetonitrile species [Au(NCMe)(L)]⁺ (Supplementary Fig. S13).

Therefore, when Au4 is dissolved in a water/acetonitrile mixture, the Au-Cl ionization is promoted, and gold-aquo [Au(OH₂)(L)]⁺ and [Au(NCMe)(L)]⁺ species are observed in the mixture. These observations are commented in the main text (page 6).

9. Regarding the sentence in Page 11 "bearing a triflimide (NTf₂) instead of a Cl counterion ([Au(PTA)]NTf₂, Au1'),"

Reviewer (1st / 2nd revision): A confusing formula with one-coordinate Au(I) and uncoordinated Ntf. Ntf is not defined in the script. Is it neutral or negatively charged? In Si section 4.6 I find the use of AgNtf₂ so I assume NTF is an anion (deprotonated NH?), in which case this is a Ag(II) complex – very unusual, strong oxidant I assume.

The Au complex would also be Au(II) as written. The formula is written with one-coordinate Au(II) and two non-coordinated NTfs.

Very confusing. These experiments must be reported in a way in which they can be repeated. As far as I can see there is no mention of the source of the NTf compounds.

Reply: The use of the term “triflimide” versus “bis(trifluoromethanesulfonyl)imidate or bistriflimidate”, might have confused the referee.

Bistriflimidate gold complexes of this type $[\text{Au}(\text{NTf}_2)(\text{L})]$, first reported by Gagosz in 2005, have been extensively used in gold catalysis, in organic solvents. These authors even provided crystallographic characterization (pubs.acs.org/doi/abs/10.1021/ol0515917). Therefore, **Au1'** is just a gold(I) complex with a neutral phosphine ligand (PTA) and an anionic ligand ($[\text{NTf}_2]^- = \text{N}(\text{SO}_2\text{CF}_3)_2$) coordinated to gold. To avoid confusions, we modified the way in which we wrote the complex in the manuscript to $[\text{Au}(\text{NTf}_2)(\text{PTA})]$. Therefore, **Au1'** is not an Au(II) complex, but rather an Au(I) species; and AgNTf_2 [CAS Number 189114-61-2] is also a Ag(I) salt.

We have implemented a new procedure for the synthesis of **Au1'** (described in the supplementary information), which is more reliable than the previous one, for large scale synthesis.

10. Reviewer (2nd revision): I asked previously: “What catalyst is used in this reaction. It says on the arrow $[\text{AuPPh}_3]$ is the metal complex- impossible. If the NTf is coordinated it should be inside the Au bracket. Is there an x-ray structure of this complex?”

The response below is unacceptable.

First reply: The gold complex used for the synthesis of some of the substrates of this work is commercial available from Sigma Aldrich as [Bis(trifluoromethanesulfonyl)imidate](triphenylphosphine)gold(I) (2:1) toluene adduct with SKU 677922.

Reply: As indicated above, these kind of gold(I) complexes are quite standard in gold(I) catalysis. Nevertheless, it is true that the bistriflimidate anion $[\text{NTf}_2]^-$ should be represented inside the brackets, as it is coordinated to gold. Therefore, we modified the formula accordingly. Additionally, the figure has been changed on the arrow. We are using an ALDRICH commercial available reagent (<https://www.sigmaaldrich.com/catalog/product/ALDRICH/677922?lang=en®ion=GB>), and it can be easily prepared as described in pubs.acs.org/doi/abs/10.1021/ol0515917. The x-Ray is provided in the reference above.

11. Regarding the sentence in Page 12: “These results confirm that gold species can effectively enter living cells, but also point out that the entrance depends on the characteristics of the ligand”

Reviewer (2nd revision): And on the redox behaviour. The Au(II) complexes are likely to be reduced in the culture medium.

Characteristics (spelling)

Reply: This comment is probably due to the interpretation that gold has two NTf ligands. However, the $[\text{NTf}_2]^-$ (bistriflimidate) group is just one ligand, possessing two **trifluoromethanesulfonyl** groups, and it's represented as NTf_2 . For this reason, the gold complex we are employing is a gold(I) complex.

The word “characteristics” has been corrected.

12. **Reviewer (2nd revision):** As drawn in Fig S8, M is a +1 ion so M+H would carry a +2 charge

Reply: Actually, our initial assignment was not properly written in the figure because the ESI equipment was slightly decalibrated during those experiments. After recalibration, we reinjected the samples and now the *m/z* values are in full agreement with those predicted, being both of them +1 ions (see new figure S3).

Comments and answers to Reviewer 3

Reviewer: P3: “ease access” needs to be changed to “easy access”;

Reply: ease access changed to easy access.

Reviewer: P3: “catalytic network” should be removed since the two processes are not connected and therefore are not part of a “network” (in fact, orthogonal as described in the paper);

Reply: catalytic network has been changed to “mutually orthogonal, metal-promoted transformations”.

Reviewer: P4: Figure 1 should be changed. “first gold-catalyzed C-C bond forming reaction in water” needs to be modified as technically “gold-catalyzed C-C bond forming reactions in water” have been previously reported (e.g. see Wei, C., Li, C.-J. JACS, 2003, 125, 9584); This statement should also be modified at the end of the manuscript. Furthermore, “complete chemo- and regioselectivity” should be changed to “complete chemoselectivity” as the substrates were carefully chosen to achieve regioselectivity of the transformations.

Reply: The proposed changes have been incorporated to the figure 1 and the main text. Moreover, a recently published paper dealing gold nanoparticles has also been incorporated (*Angew. Chem. Int. Ed.*, **2017**, 129, 12722).

Reviewer: P5: The authors might consider removing figure 2, as it doesn't bear any important information and only confuses the reader. On the contrary, comparison of cytotoxicities of tested gold complexes is an important piece of data, which should be presented in the main text of the paper (e.g. MTT assay), perhaps later in the text.

Reply: According to the suggestion of the referee, we have included the MTT assay in Fig. 3 of the manuscript.

Reviewer: P5: It would be good to see the comparison of ³¹P chemical shifts in acetonitrile and water for the main complex (Au1) used in the study.

Reply: These experiments have now been performed and the results are presented in detail in the Supplementary Information, Section S8). Furthermore, the results are complemented with MS data, which fully confirms the formation of aquo, acetonitrile and chloride complexes. Moreover, we have confirmed that the presence of water is necessary for the partial dissociation of the chloride.

Reviewer: Taking into account the inhibitory effect of cysteine and glutathione in *in vitro* studies it will be interesting to hear the authors' thoughts/rationale for the efficiency of the catalytic process in cells, where multiple reactive cysteines and glutathione are present, which must deactivate the catalyst.

Reply: This is an interesting issue. The experimental fact is that there is reaction in the cellular environments. This can be explained in terms of a much lower concentration and availability of reactive thiols in the intracellular sites in which the reaction is occurring.

Reviewers' comments:

Reviewer #2 (Remarks to the Author):

The authors have attempted to answer the previous criticisms, but several others remain.

"intracellular turnover has not been really investigated."
Or investigated at all. It is not clear what the active catalyst actually is.

"glycoalbumin-gold(III) complex for a propargyl ester amidation in mice;26"
Do the authors believe this paper (ref 26)? I see no evidence in ref 26 that they have characterized the catalysts as a Au(III) complex and they describe it as a dichloride complex despite working in water where hydrolysis would be expected.

"In many cases the reactions can be carried out using gold(I) chlorides, but they require the addition of chloride scavengers such as silver(I) salts to generate vacant coordination sites."
You do not always need a scavenger to stabilize a Au(I)-Cl bond. For example AuCl₂⁻ itself is quite unstable in aqueous media.
You do not generate a 'vacant' coordination site'. One-coordinate Au(I) is unknown. You replace chloride by a more labile ligand.

There is nothing unexpected about the hydrolysis behavior they observe in water.

Fig S14

Negative ion MS at top, positive ion at bottom?

Piperidine adduct might be neutral?

Aqua not aquo

Page 8

"probably because of the low aqueous solubility of the complex which hinders its water"

Fig S16 is an NMR figure. We need to know the solubility determined for example by AA or ICP-MS.

Fig S15

Peak assignments need to be added

Page 9

"Not surprisingly, the presence of excess of thiols (glutathione and cysteine) in the reaction media led to an important inhibition of the catalytic activity,"

This is important -more details are needed, not just a vague sentence. In cells GSH is ca. 2-10 mM. Do 2-8 mM millimolar concentrations of GSH completely inhibit activity?

They say in the abstract that "Key to the success of the process is the use of designed, "water-activatable" gold chloride complexes." But it seems that such aqua species are unlikely to exist inside cells.

S7.3:proper labelling of the subparts is needed

Page 12

"a control experiment with a preformed complex similar to Au1 but bearing a bistriflimidate [NTf₂]⁻ instead of a Cl⁻ counterion ([AuNTf₂(PTA)]⁺),"

I do not understand this statement. If in this formula NTf₂ is supposed to be a counteranion then it should not be inside the square brackets. On the other hand if they mean chloride is coordinated they should not refer to it as a counter anion. It is a ligand.

Page 13

"The gold complex Au1 showed a 50-fold increase of intracellular Au concentration with respect to non-treated blank cells,"

This is amazing. I would not expect there to be any gold in the non-treated cells and therefore the the ratio should be much much higher.

Nowhere do I see a catalytic cycle. Figure 2 has undefined [Au] as the catalyst ad Fig 5 has complex 1 but that is unlikely to exist in cells in the presence of a large excess of GSH.

The generation of fluorescent species in cells at relatively high metal concentrations is of limited value.

More work is needed on the speciation of gold at micromolar concentrations in cells but none of the available methods has been attempted here.

Reviewer #3 (Remarks to the Author):

After reviewing the changes incorporated by Mascareñas et al. I am satisfied with the added mechanistic details with regard to the nature of organometallic species in solution and believe that the manuscript meets the criteria for publication in Nature Communications after some minor changes. My major concern is still the ability of cysteine and GSH to inhibit the reaction (which technically makes it less bioorthogonal than stated), which is further reflected by significantly diminished yields in cell lysates, however this issue can be addressed at a later point.

Main text:

p.9 line 195: "These promising bioorthogonal assays prompted us to study the transformation in biological media of diverse complexity" Considering the inhibitory effects of cysteine and GSH, this overly optimistic sentence needs to be modified.

p.10 line 200: for better logic, I would list the experiments in the order of increasing complexity and mention the BSA experiment before the one with cell lysates. I would also add a potential explanation for lower lysate reactivity (e.g. potentially due to the presence of GSH and hyperreactive cysteines [Nature 2010, 468, 790]).

References 3-14 need to be updated. Incorporation of several general more recent reviews would be beneficial for the readers and for the appropriate description of the existing body of work. E.g., Angew. Chem. Int. Ed. 2017, 56, 1521; Pure Appl. Chem. 2017, 89, 1619; Curr. Opin. Chem. Biol. 2014, 21, 128; Chem. Soc. Rev. 2014, 43, 6511.

p.3 line 54: change "might not be not strictly needed" to "might not be strictly needed"

p.4 line 83: "fluorescent-inducing" change to "fluorescence-inducing"

p.5 line 97: the new text is too extensive for the main text of the paper and should therefore be shortened to highlight the results, while the experiment set up can be in the SI. E.g., "To shed some light on this water-promoted activation process, we performed a series of NMR and ESI-MS studies (see Supplementary Information, section S8), which confirmed that water promotes the ionization of the Au-Cl bond and thus drives the complexation of the reactants to the gold(I) complex, which eventually allows to initiate the catalysis."

p.8 line 172: change "see Supplementary Fig. S17" to "see Supplementary Fig. S17 and Fig. 3a"

Supporting Information:

"Monitorization" change to "monitoring"

POINT BY POINT reply to the Reviewers Comments.

Comments and answers to Reviewer 2

1.- *“intracellular turnover has not been really investigated.” Or investigated at all. It is not clear what the active catalyst actually is.*

Reply: Honestly, we had a hard time trying to understand the purpose of these comments, and whether the reviewer wanted us to do some exploration in this matter.

We had introduced that sentence about intracellular turnover in the introduction of the manuscript just as a general comment (page 2).

Finding out the real active catalysts inside the cell is almost impossible; even if we were able to isolate gold complexes from the cell extracts, we could not say anything about the real catalytic species. In general, investigating molecular details of intracellular reactions is not yet within reach. **What is clear is the structure of the gold species that we add to cells, and their “in vitro” performance, including mechanistic aspects.**

Regarding to the turnover in our reactions, we have very good turnovers in water and cellular media (DMEM), and some turnover in cell lysates (data included in the manuscript). However, obtaining reliably quantitative data about intracellular turnover is extremely difficult, owing to the requirement of working with millions of living cells and of using experimental protocols (extractions, washing, cell counting, etc..) that can introduce important errors. In addition, the quantitative results should be taken with caution as we don't know how much substrate is internalized or which proportion of the intracellular gold is part of active complexes.

Nevertheless, interpreting that the reviewer might want us to do some investigation in this matter, we carried out some experiments to gain further information on the intracellular reactivity. Using ICP-MS to measure the intracellular amount of gold, and measuring the fluorescence intensity of cellular extracts after the reaction (using in vitro fluorescence calibration curves), we could estimate an average turnover that is slightly above one. However, we want to stress that, for the reasons indicated above, this is just an approximation.

Anyway, we decided to include these experiments as additional information in the last section of the supplementary information, but, importantly, we have added a “caution” call to be sure that potential readers understand the context of the experiments and do not overinterpret the results.

2.- *“glycoalbumin-gold(III) complex for a propargyl ester amidation in mice,²⁶”*

Do the authors believe this paper (ref 26)? I see no evidence in ref 26 that they have characterized the catalysts as a Au(III) complex and they describe it as a dichloride complex despite working in water where hydrolysis would be expected.

Reply: I think we cannot enter into judging work that has been published in peer reviewed journals.

3.- *“In many cases the reactions can be carried out using gold(I) chlorides, but they require the addition of chloride scavengers such as silver(I) salts to generate vacant coordination sites.”*

You do not always need a scavenger to labilize a Au(I)-Cl bond. For example AuCl₂⁻ itself is quite unstable in aqueous media. You do not generate a ‘vacant’ coordination site’. One-coordinate Au(I) is unknown. You replace chloride by a more labile ligand.

Reply: We have changed the sentence to “In many cases the reactions can be carried out using gold(I) chlorides, but they require the addition of chloride scavengers such as silver(I) salts to replace chloride by a more labile ligand.” (page 3)

4.-*There is nothing unexpected about the hydrolysis behavior they observe in water.*

Reply: Well, the relevance is not the hydrolysis. The relevance is the demonstration that in presence of water there is a partial formation of aqua species, which suggests that water is mediating the labilization of the chloride ligand, something critical for reactivity (without water, there is not reaction).

We have slightly revised the sentences in this topic, to avoid confusion on the role of water (highlighted in yellow).

5.-*Fig S14*

Negative ion MS at top, positive ion at bottom? Piperidine adduct might be neutral?

Reply: The adducts with the coordination of the piperidine are charged (+1).

6.-*Aqua not aquo*

Reply: We believe that both words can be used; but have changed to aqua in the manuscript, as suggested by the reviewer.

7.-*Page 8 “probably because of the low aqueous solubility of the complex which hinders its water”*

Fig S16 is an NMR figure. We need to know the solubility determined for example by AA or ICP-MS.

Reply: We are a little puzzled by the insistence in this matter. This is lateral information that is not relevant in the context of the work.

We could do many alternative experiments, but we honestly consider that NMR data are completely illustrative. Solubility of **Au6** is extremely poor since it is not detected at all by NMR. An absolute value will not modify this conclusion, neither bring new ones.

8.-*Fig S15*

Peak assignments need to be added

Reply: revised

9.-*Page 9: “Not surprisingly, the presence of excess of thiols (glutathione and cysteine) in the reaction media led to an important inhibition of the catalytic activity,”*

This is important -more details are needed, not just a vague sentence. In cells GSH is ca. 2-10 mM. Do 2-8 mM millimolar concentrations of GSH completely inhibit activity?

Reply: When in vitro studies were carried out with cell lysates, which maintain the same levels of GSH as in living cells, we observe after 24h a 27% of yield, which confirms that the reaction tolerates moderate levels of thiols (page 9). Inside cells is likely that thiols are not as freely available as in solution, and therefore the inhibitory role can be more compromised. The experimental fact is that inside cells we do observe activity.

We have slightly revised the text highlighted in page 9.

10.-*They say in the abstract that “Key to the success of the process is the use of designed, “water-activatable” gold chloride complexes.” But it seems that such aqua species are unlikely to exist inside cells.*

Reply: As indicated previously, it is not possible to know the mechanism of the reaction inside cells; however, the in vitro studies confirm that the reactivity of the gold chloride requires water, and therefore it is reasonable to deduce that also inside cells water is critical to labilize the chloride and trigger the reactivity. Therefore, we speak about “water activatable” process; we never state that aqua species exist inside cells.

11.-S7.3: proper labelling of the subparts is needed

Reply: revised

12.-Page 12: “a control experiment with a preformed complex similar to Au1 but bearing a bistriflimidate [NTf2]- instead of a Cl- counterion ([AuNTf2(PTA)],”

I do not understand this statement. If in this formula NTf2 is supposed to be a counteranion then it should not be inside the square brackets. On the other hand if they mean chloride is coordinated they should not refer to it as a counter anion. It is a ligand.

Reply: We apologize if there was any misunderstanding, but the situation is clear: NTf₂ is not a counterion, but a ligand, just like the chloride. Indeed a comparison of the X-ray data available in the literature for [AuCl(PPh₃)] (Baezinger N. C. et al. *Acta Crystallographica, Section B: Struct. Crystallogr. Cryst. Chem.*, **1976**, 32, 962), [AuNTf₂(PPh₃)] (Gagosz et al. *Organic Letters* **2005** 7, 4133), and even for [AuI(PPh₃)] (*Acta Chemica Scandinava A* 41, **1987**, 173) shows that the Au-N distance in [AuNTf₂(PPh₃)] is not only shorter than that of the iodide complex, but also shorter than the Au-Cl analog in [AuCl(PPh₃)] [Au-Cl: 2.278 Å; Au-I 2.55 Å; and Au-N 2.102 Å].

We have changed the word “counterion” for “ligand” in the case of the chloride complex (page 11).

13.-Page 13: “The gold complex Au1 showed a 50-fold increase of intracellular Au concentration with respect to non-treated blank cells,”

This is amazing. I would not expect there to be any gold in the non-treated cells and therefore the the ratio should be much much higher.

Reply: To avoid any misunderstanding we have changed the sentence to say “With complex **Au1** we observed a gold concentration of 195.89 ng/10⁶ cells” (page 12).

14.-Nowhere do I see a catalytic cycle. Figure 2 has undefined [Au] as the catalyst ad Fig 5 has complex 1 but that is unlikely to exist in cells in the presence of a large excess of GSH.

Reply: The mechanism behind this type of Au-promoted cyclizations has been previously investigated (see for instance Kim *et al*, *Org. Lett.*, **2010**, 12, 932). In figure 1, [Au] is a general term to refer to different gold species, and in figure 5 we indicate the transformations that take place and the “precatalysts” that we are sure we add. The real reactive species inside cells is impossible to know.

15.-The generation of fluorescent species in cells at relatively high metal concentrations is of limited value.

Reply: Of course, the generation of fluorescent product doesn't intend to be a practical application. The work described in our article is conceptual, and a proof of principle stage; and the best current way of monitoring the intracellular activity is based on fluorescence microscopy, which is also a standard technique in cell biology. Overall, we are opening a “new and interesting area for future development”, as remarked by Reviewer 1 in its first revision, and applications are expected in the future.

16.-More work is needed on the speciation of gold at micromolar concentrations in cells but none of the available methods has been attempted here.

Reply: As already commented, knowing the fate of the gold species inside cells is far from trivial, and does not necessarily bring any meaningful information. In our work we demonstrate that using the indicated gold complexes it is possible to promote programmed carbocyclizations inside cells, and in this way we are contributing to further establishing this research field.

Comments and answers to Reviewer 3

1.-p.9 line 195: "These promising bioorthogonal assays prompted us to study the transformation in biological media of diverse complexity" Considering the inhibitory effects of cysteine and GSH, this overly optimistic sentence needs to be modified.

Reply: The sentence has been slightly changed to "This reasonable biorthogonal profile prompted us..." (page 8)

2.-p.10 line 200: for better logic, I would list the experiments in the order of increasing complexity and mention the BSA experiment before the one with cell lysates. I would also add a potential explanation for lower lysate reactivity (e.g. potentially due to the presence of GSH and hyperreactive cysteines [Nature 2010, 468, 790]).

Reply: revised

3.-References 3-14 need to be updated. Incorporation of several general more recent reviews would be beneficial for the readers and for the appropriate description of the existing body of work. E.g., *Angew. Chem. Int. Ed.* 2017, 56, 1521; *Pure Appl. Chem.* 2017, 89, 1619; *Curr. Opin. Chem. Biol.* 2014, 21, 128; *Chem. Soc. Rev.* 2014, 43, 6511.

Reply: The references 3-14 have been updated but we could not include all the suggested references owing to the editorial rules that restrict the number to 50.

Angew. Chem. Int. Ed. 2017, 56, 1521 does not exist; *Pure Appl. Chem.* 2017, 89, 1619 is older than references 1 and 2; and *Curr. Opin. Chem. Biol.* 2014, 21, 128 only cover Pd chemistry.

4.-p.3 line 54: change "might not be not strictly needed" to "might not be strictly needed"

Reply: revised

5.-p.4 line 83: "fluorescent-inducing" change to "fluorescence-inducing"

Reply: revised

6.-p.5 line 97: the new text is too extensive for the main text of the paper and should therefore be shortened to highlight the results, while the experiment set up can be in the SI. E.g., "To shed some light on this water-promoted activation process, we performed a series of NMR and ESI-MS studies (see Supplementary Information, section S8), which confirmed that water promotes the ionization of the Au-Cl bond and thus drives the complexation of the reactants to the gold(I) complex, which eventually allows to initiate the catalysis."

Reply: According to the suggestion of the referee, the main text has been shortened (page 5), and this paragraph has been added to the Supplementary Information, section 8.

7.-p.8 line 172: change “see Supplementary Fig. S17” to “see Supplementary Fig. S17 and Fig. 3a”

Reply: revised

8.-“Monitorization” change to “monitoring”

Reply: revised

REVIEWERS' COMMENTS:

Reviewer #3 (Remarks to the Author):

I believe the revised version of the manuscript entitled "Concurrent, orthogonal gold(I) and ruthenium(II) catalysis inside living cells" by Prof Mascareñas and colleagues addresses the points raised in the previous round of reviews and meets the criteria for publication in Nature Communications.

The suggested ACIE reference for the introduction is Angew. Chem., Int. Ed. 2017, 56, 10644. The previously mentioned references to several reviews should be added to the reference list as they are complementary to the already cited reviews and include the currently listed references 3-11, some of which can be exchanged.

POINT BY POINT reply to the Reviewers Comments

Comment and answer to Reviewer 3

Reviewer- "The suggested ACIE reference for the introduction is *Angew. Chem., Int. Ed.* 2017, 56, 10644. The previously mentioned references to several reviews should be added to the reference list as they are complementary to the already cited reviews and include the currently listed references 3-11, some of which can be exchanged"

Reply: According to the reviewer's suggestion, we have been added the mentioned reference, and those mentioned in the previous revision rounds, that is: *Pure Appl. Chem.* **2017**, 89, 1619; *Curr. Opin. Chem. Biol.* **2014**, 21, 128 and *Chem. Soc. Rev.* **2014**, 43, 6511.